# Generalized One-shot Domain Adaptation of Generative Adversarial Networks

**Zicheng Zhang**[1]* **Yinglu Liu**[2] **Congying Han**[1]† **Tiande Guo**[1] **Ting Yao**[2] **Tao Mei**[2]

[1]University of Chinese Academy of Sciences    [2]JD AI Research

zhangzicheng19@mails.ucas.ac.cn    liuyinglu1@jd.com    hancy@ucas.ac.cn
tdguo@ucas.ac.cn    tingyao.ustc@gmail.com    tmei@live.com

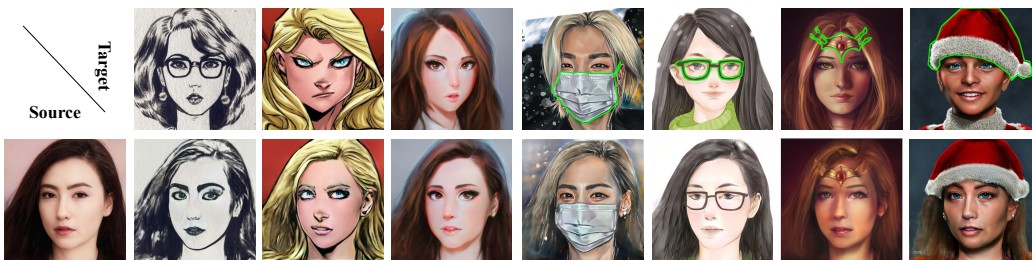

Figure 1: Illustrations of the generalized one-shot GAN adaptation task. Target images with the corresponding binary entity masks are provided to adapt the pre-trained GAN to the domain of similar style and entity as target images. The first three masks are all zeros and the others have value 1 in the areas bounded by green lines. Our adapted model is of strong cross-domain correspondence to realize both style and entity transfer.

## Abstract

The adaptation of a Generative Adversarial Network (GAN) aims to transfer a pre-trained GAN to a target domain with limited training data. In this paper, we focus on the one-shot case, which is more challenging and rarely explored in previous works. We consider that the adaptation from a source domain to a target domain can be decoupled into two parts: the transfer of global style like texture and color, and the emergence of new entities that do not belong to the source domain. While previous works mainly focus on style transfer, we propose a novel and concise framework to address the *generalized one-shot adaptation* task for both style and entity transfer, in which a reference image and its binary entity mask are provided. Our core idea is to constrain the gap between the internal distributions of the reference and syntheses by sliced Wasserstein distance. To better achieve it, style fixation is used at first to roughly obtain the exemplary style, and an auxiliary network is introduced to the generator to disentangle entity and style transfer. Besides, to realize cross-domain correspondence, we propose the variational Laplacian regularization to constrain the smoothness of the adapted generator. Both quantitative and qualitative experiments demonstrate the effectiveness of our method in various scenarios. Code is available at `https://github.com/zhangzc21/Generalized-One-shot-GAN-Adaptation`.

## 1 Introduction

Benefiting from numerous excellent pre-trained GANs (*e.g.,* StyleGAN [16] and BigGAN [6]), GAN adaptation has become an important research topic, which leverages the knowledge of GANs

---

*Work done during an internship at JD AI Research.
†Corresponding author

36th Conference on Neural Information Processing Systems (NeurIPS 2022).

pre-trained on large-scale datasets to mitigate the lack of data and speed up the training on a new domain. The adaptation can be divided into few-shot, one-shot and zero-shot [11] cases. The few-shot (usually $\geq 10$) case has been extensively studied in the past years [42, 41, 26, 22]. The latest works [29, 43, 46] seek the cross-domain correspondence that keeps the shapes or contents of syntheses before and after adaptation invariant. Recently, the more challenging one-shot case has been explored by several works [47, 55, 20], which mainly focus on adapting GANs to the target domain of similar style as the given image. By utilizing GAN inversion [45], the adapted models can be applied to some creative tasks like image style transfer and manipulation [55].

However, we perceive there exist limitations in previous task settings [47, 55, 20], due to the fact that an exemplar contains rich information more than artistic style including color and texture. As shown in Fig. 1, the source domain refers to natural faces, and the exemplars of target domains are artistic portraits with some *entities* (*e.g.,* hat and accessories). Previous works only concern the artistic style transfer while the entities are neglected. This may lead to two problems: 1) In some cases, the entities considered as an important part of domain feature, should be transferred with the style simultaneously. 2) The entities of large size, like the mask in Fig. 1, easily bias the style extraction of face region and cause undesired artifacts in adapted syntheses. Therefore, we perfect the task settings with the aid of an extra binary mask, which labels the entities we are interested in and helps to define the target domain more exactly and flexibly. We name the new task *generalized one-shot GAN adaptation*. The scenarios in previous works can be taken as the special case when the binary mask is full of zeros. We believe the exploration of the novel task is quite meaningful, which not only serves to artistic creation for more general and complex scenarios, but also facilitates further discovery and utilization of the knowledge in pre-trained models.

To tackle the new adaptation problem, we propose a concise and effective adaptation framework for StyleGAN [17], which is a frequent base model in previous works [47, 55, 20]. Firstly, we modify the architecture of the original generator to *decouple the adaptation of style and entity*, where the new generator comprises an additional auxiliary network to facilitate the generation of entities, and the original generator is dedicated to generating stylized images. Secondly, unlike previous works using CLIP similarity [55] or GAN loss [47] to learn the domain knowledge, our framework directly *minimizes the divergence of the internal distributions* of the exemplar and syntheses by sliced Wasserstein distance [5] for both style and entity transfer. Combined with the *style fixation* that attaches all syntheses with exemplary style by the transformation in latent space, our framework is efficient and potent enough to obtain compelling results. Thirdly, inspired by the cross-domain consistency [29] and classic manifold learning [12, 2], we propose the *variant Laplacian regularization* that smooths the network changes before and after adaptation to preserve the geometric structure of the source generative manifold, and prevent the content distortion during training. We conduct extensive experiments on various references with and without entities. The results show that our framework can fully exploit the cross-domain correspondence and achieve high transfer quality. Moreover, the adapted model can handily perform various image manipulation tasks.

## 2 Related works

GAN adaptation leverages the knowledge stored in pre-trained GANs to mitigate the lack of data and speed up the training on a new domain. [42] is the first to study and evaluate different adapting strategies. It shows that transferring both pre-trained generator and discriminator can improve the convergence time and the quality of the generated images. Since then, more and more adapting strategies [28, 41, 32, 53] are proposed for various GAN architectures [14, 6, 25]. As StyleGAN [16] and its variants [17, 15] continue to make breakthrough progress in the generation of various classes of data, designing the adapted strategies of StyleGAN has gained a lot and sustained attention. Specifically, [30, 19] have explored the zero-shot GAN adaptation, which transfers StyleGAN into the target domain defined by the text prompts with the help of CLIP [31]. [22, 29, 21, 43, 46, 18] study to transfer the StyleGAN to the domain defined by the given few-shot images.

Among the above works, our work has a close relation to FSGA [29], which proposes the cross-domain consistency (CDC) loss to maintain the diversity of the source generator, and makes the transferred and source samples have a corresponding relation. Although CDC loss is ineffective for FSGA when the training data is only one-shot, its mechanism inspires us to consider a more effective regularization to maintain the relation, which will be discussed in Sec. 4.4.

Recently, [55, 20, 9, 47] have explored the one-shot adaptation task. [55, 20] align the adapted samples and reference image in the CLIP feature space. [9] learns a style mapper by constructing a substantial paired dataset. [47] introduces the extra latent mapper and classifier to the original GAN. In contrast, we focus on both style and entity adaptation which has never been studied before. And we prove that under the decoupling of style and entity, aligning the internal distributions with the proper regularization is effective for GAN adaptation.

## 3    Preliminaries

**StyleGAN**    A canonical StyleGAN [17] can be summarized in two parts: Firstly, given a noise distribution $P(z)$, a mapping network transforms the noise from space $\mathcal{Z} = \{z|P(z) \neq 0\} \subseteq \mathbb{R}^{1 \times 512}$ into the replicated latent space $\mathcal{W} = \{[F(z); \dots; F(z)]|z \in \mathcal{Z}\} \subseteq \mathbb{R}^{18 \times 512}$, where $F$ is a fully-connected network. Secondly, a synthesis network $G$ composed of convolutional blocks transfers $\mathcal{W}$ space into image space $\mathcal{M} = \{G(w)|w \in \mathcal{W}\} \subseteq \mathbb{R}^{H \times W \times 3}$. Another important concept is the extended latent space $\mathcal{W}^+ = \{[F(z_1); \dots; F(z_{18})]|z_i \in \mathcal{Z}\}$. For $\forall\, w_1, w_2 \in \mathcal{W}$, the latent code manipulation merges their information by the linear combination $diag(\alpha_1)w_1 + diag(\alpha_2)w_2 \in \mathcal{W}^+$, $\alpha_1, \alpha_2 \in \mathbb{R}^{18}$. Empirically, $\mathcal{W}^+$ space is inferior to $\mathcal{W}$ in generation fidelity and editability, but superior in GAN inversion that finds a code $w_{ref}$ to reconstruct a reference $y_{ref} \in \mathbb{R}^{H \times W \times 3}$. In this paper we only focus on tuning $G$, and fix $F$ for keeping the distribution of source latent spaces.

**Manifold learning**    Considering the sample matrix $X = [x_1^T; \dots; x_n^T]$ from source manifold $\mathcal{M}_s$, $W = [w_{i,j}]_{i,j=1}^n$ is the weight matrix, where $w_{i,j}$ is the weight of $x_i$ and $x_j$ usually defined as $e^{-\|x_i - x_j\|^2/\sigma}$. Given a task-specific function $f$ and $y = f(x)$, manifold learning requires the transformed samples $Y = [y_1^T; \dots; y_n^T]$ on the new manifold $\mathcal{M}_t$ should preserve the geometric structure of $\mathcal{M}_s$, which means the data nearby on $\mathcal{M}_s$ are also nearby on $\mathcal{M}_t$, and vice versa. The requirement is mostly realized by minimizing the manifold regularization term [4] $\int_{\mathcal{M}_s} \|\nabla_{\mathcal{M}_s} y\|^2 dP(x)$, $\nabla_{\mathcal{M}_s}$ is the gradient along manifold, and $P(x)$ is the probability measure. If $\mathcal{M}_t$ is a submanifold of $\mathbb{R}$, the former is equalized with $\int_{\mathcal{M}_s} y \Delta_{\mathcal{M}_s} y dP(x)$ by Stokes theorem. $\Delta$ called the Laplace–Beltrami operator, is a key geometric object in the Riemannian manifold, thus the integral is also called Laplacian regularization. In practice, it is estimated by the discrete form $2tr(Y^T L Y) = \sum_{i,j} w_{i,j} \|y_i - y_j\|^2$, $L = D - W$ is known as the graph Laplacian operator, $D$ is the degree matrix where the diagonal element $d_{i,i} = \sum_{j=1}^n w_{i,j}$ and other elements are zero. The convergence between $L$ and $\Delta$ can be found in [3]. Intuitively, the regularization encourages the local smoothness of $f$. If data distribute densely in an area, the function should be smooth to avoid local oscillations. If $x_i$ and $x_j$ are close, in the discrete form $w_{i,j}$ will impose a strong penalty to the distance between $y_i$ and $y_j$.

**Task definition**    The generalized one-shot adaptation can be formally described as: Given a reference $y_{ref}$, a mask $m_{ref} \in \{0, 1\}^{H \times W}$ is provided to label the location of entity contained in $y_{ref}$. The target domain $\mathcal{T}$ is defined to contain images of similar style and entities as $y_{ref}$. With the knowledge stored in a generative model $G_s$ pre-trained on source domain $\mathcal{S}$, a generative model $G_t$ is learned from $y_{ref}$ and $m_{ref}$ to generate diverse images belonging to domain $\mathcal{T}$. Note that we can set $m_{ref} = 0$ when there is no interested entities existed in $y_{ref}$. Moreover, the adaptation should satisfy the cross-domain correspondence. *i.e.*, $\forall\, w \in \mathcal{W}$, $G_s(w)$ and $G_t(w)$ should have similar shape or content in visual. With the correspondence, adapted models can be applied for not only synthesis, but also the transfer and manipulation of source images.

## 4    Methodology

### 4.1    Overall

Our framework for tackling the generalized one-shot GAN adaptation task is illustrated in Fig. 2. For the sake of description, in the following part we will take the StyleGAN pre-trained on FFHQ as an example to describe each component of the framework in detail. We will demonstrate our framework can be applied to various domains in the experiments.

**Network architecture**    Different from previous works [55, 20] that directly adapt the source generator $G_s$, our target generator $G_t$ (as Fig. 2 shows) consists of a generator $\hat{G}_t$ inherited from $G_s$, and an auxiliary network ($aux$) trained from scratch. We are inspired by the fact that $G_s$ can

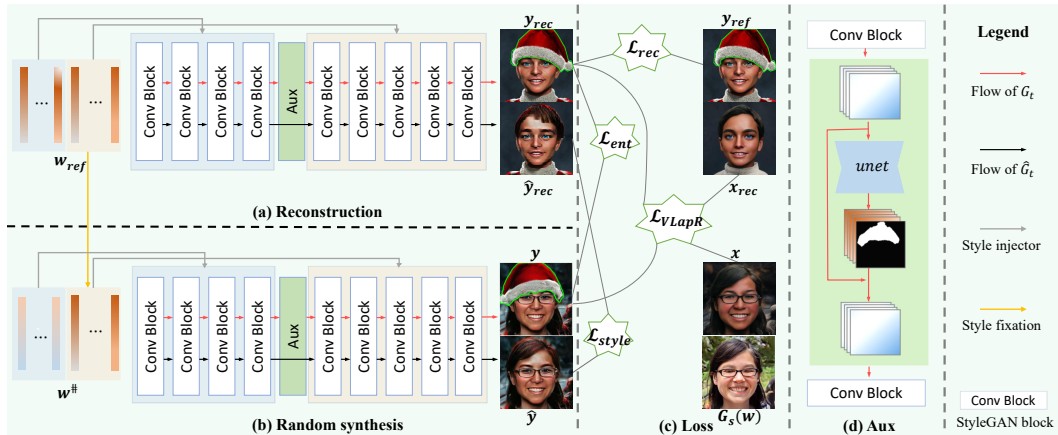

Figure 2: The proposed framework for generalized one-shot GAN adaptation. The convolutional blocks are inherited from the original StyleGAN [17]. The black arrows represent the flow of $\hat{G}_t$, and the red arrows represent the flow of $G_t$. For the details please refer to Sec. 4.1.

approximate arbitrary real faces but cannot handle most entities, due to that $G_s$ captures the prevalent elements (clear natural faces) of FFHQ rather than the tails (faces with entities like hats or ornaments) of the distribution [49, 27]. The design makes $aux$ serve to entities, and $\hat{G}_t$ just focus on stylizing clear face to benefit from the prior knowledge stored in $G_s$. As shown in Fig. 2(d), for each $w$, its feature map generated by the convolutional block of StyleGAN is fed into $aux$, where a UNet [33] predicts the feature map $f_{ent}$ of entity and the mask $m$ that exactly labels the position of the entity, then they are merged with $f_{in}$ by Eq. (1) to get the feature map of face covered with entity.

$$aux(f_{in}) = (1 - m) \odot f_{in} + m \odot f_{ent}, \tag{1}$$

$\odot$ denotes the Hadamard product. We hope the shape of the entity is only influenced by the content of synthesis, hence we place $aux$ after the fourth block of StyleGAN, where the $32 \times 32$ resolution $f_{in}$ contains the most content information and the least style information of synthesis (see Sec. 4.2). Because the global style and local entity are usually independent, the disentangled design also prevents the style of faces from being contaminated by that of entities (see Sec. 4.3).

**Training process** Our training process can be divided into three main parts: ① Style fixation and exemplar reconstruction. In Fig. 2(a), we get the latent code $w_{ref} = [w^c_{ref}; w^s_{ref}]$ of $y_{ref}$ by the GAN inversion technique [39], and train $G_t$ to reconstruct $y_{ref}$ with reconstruction loss $\mathcal{L}_{rec}$. In Fig. 2(b), each $w$ will be transformed into the style-fixed code $w^\sharp = [w^c; w^s_{ref}]$ to roughly obtain the exemplary style. ② Internal distribution learning. We minimize the divergence of internal patch distributions between syntheses and exemplar for style and entity transfer. Instead of using GAN loss that is hard to optimize, we take sliced Wasserstein distance (SWD) [5] as style loss $\mathcal{L}_{style}$ and entity loss $\mathcal{L}_{ent}$ to learn internal distribution efficiently. ③ Manifold Regularization. To suppress the content distortion during training, we propose the variational Laplacian regularization $\mathcal{L}_{VlapR}$ to smooth the change from $G_s$ to $G_t$ by preserving the geometric structure of the source manifold. As shown in Fig. 2(c), the overall loss is

$$\min_{G_t} \lambda_1 \mathcal{L}_{rec} + \lambda_2 \mathcal{L}_{style} + \lambda_3 \mathcal{L}_{ent} + \lambda_4 \mathcal{L}_{VLapR} \tag{2}$$

The details of training process will be illustrated and discussed in the following sections. For convenience of description and self-consistency, some abbreviations have been taken: $y = G_t(w^\sharp)$, $\hat{y} = \hat{G}_t(w^\sharp)$, $y_{rec} = G_t(w_{ref})$, $\hat{y}_{rec} = \hat{G}_t(w_{ref})$, $x = G_s(w^\sharp)$, $x_{rec} = G_s(w_{ref})$.

### 4.2 Style fixation and exemplar reconstruction

It is well known that StyleGAN is equipped with the disentangled latent spaces $\mathcal{W}$ and $\mathcal{W}^+$ [1], where the latter vectors of code mainly determine the style of synthesis whereas the earlier vectors determine the coarse-structure or content of the synthesis. Taking advantage of this trait, we first roughly transfer the exemplary style to other syntheses by fixing the style code. The exemplar $y_{ref}$ is

fed into the pre-trained GAN inversion encoder e4e [39] to get the latent code $\boldsymbol{w}_{ref} = [\boldsymbol{w}_{ref}^c; \boldsymbol{w}_{ref}^s]$, where $\boldsymbol{w}_{ref}^c \in \mathbb{R}^{l \times 512}$ and $\boldsymbol{w}_{ref}^s \in \mathbb{R}^{(18-l) \times 512}$ encode content and style information of exemplar respectively. For each $\boldsymbol{w} \in \mathcal{W}$, it will be transformed to $\boldsymbol{w}^\sharp$ before fed into networks,

$$\boldsymbol{w}^\sharp = diag(\boldsymbol{\alpha})\boldsymbol{w} + diag(\mathbf{1} - \boldsymbol{\alpha})\boldsymbol{w}_{ref}, \; \alpha_i = \mathbb{1}_{i<=l}(i), \; i = 1, \ldots, 18. \tag{3}$$

The content part of $\boldsymbol{w}$ is reserved and style part is replaced by $\boldsymbol{w}_{ref}^s$, and we denote the new space as $\mathcal{W}^\sharp$. Because style fixation should not change the content of the original synthesis, by using the pre-trained Arcface model [10] to compare the identity similarity before and after style fixation, we find setting $l = 8$ is acceptable which gets $65\%$ average similarity and obtains the adequate exemplary style in visual. In MTG [55], style mixing assists as a post-processing step to improve the style quality, but the experiments (Fig. 3) show that the style of syntheses is underfitting and not consistent. In our framework, it is used as a pre-processing step to facilitate the following learning.

Because $\boldsymbol{x}_{rec}$ is just the projection of $\boldsymbol{y}_{ref}$ on the source domain, there often exists an obvious distinction between them in visual as shown in Fig. 2. Hence we adopt the reconstruction loss $\mathcal{L}_{rec}$ to narrow the gap,

$$\mathcal{L}_{rec} = dssim(\boldsymbol{y}_{rec}, \boldsymbol{y}_{ref}) + lpips(\boldsymbol{y}_{rec}, \boldsymbol{y}_{ref}) + \lambda_5 mse(\boldsymbol{m}_{rec}^\uparrow, \boldsymbol{m}_{ref}). \tag{4}$$

$dssim$ denotes the negative structural similarity metric [44] and $lpips$ is the perceptual loss [50], $\boldsymbol{m}_{rec}^\uparrow$ is the upsampled reconstructed mask from $aux$. It is worth noting $\mathcal{L}_{rec}$ acts on $G_t$ rather than the vanilla architecture $\hat{G}_t$. When $\boldsymbol{m}_{ref} = \mathbf{0}$, $aux$ does not work and $G_t$ is equivalent to $\hat{G}_t$.

### 4.3   Internal distribution learning

Recent works about internal learning [35, 37, 51, 52] prove that the internal patch distribution of a single image contains rich meaning. Minimizing the divergence of internal distributions works out many tasks like image generation, style transfer, and super-resolution. It is usually achieved by the adversarial patch discriminator [13]. However, it is well known that the training of GAN is time-consuming, unstable, and requires large GPU memory. Previous few-shot GAN adaptation works [22, 29] using GAN loss almost have the serious model collapse phenomenon. We find that slice Wasserstein distance ($SWD$) [5] can lead to the same destination but with greater efficiency. For two tensor $\boldsymbol{A}, \boldsymbol{B} \in \mathbb{R}^{H \times W \times d}$, the $SWD$ of their empirical internal distributions is defined as

$$SWD(\boldsymbol{A}, \boldsymbol{B}) = \int_{\mathbf{S}^d} \|sort(proj(\boldsymbol{A}, \boldsymbol{\theta})) - sort(proj(\boldsymbol{B}, \boldsymbol{\theta}))\|^2 \, d\boldsymbol{\theta}, \tag{5}$$

where $\mathbf{S}^d = \{\boldsymbol{\theta} \in \mathbb{R}^d : \|\boldsymbol{\theta}\| = 1\}$, $proj : \mathbb{R}^{H \times W \times d} \to \mathbb{R}^{H \times W}$ that projects each pixel from the $d$ channels to a scalar by $\boldsymbol{\theta}$, and $sort$ denotes the sort operator ordering all values. $SWD$ can be easily implemented by convolution of a randomized $1 \times 1$ kernel along with a quick-sort algorithm.

To realize style and entity transfer, we use $SWD$ to minimize the divergence of the empirical internal distribution of the syntheses and reference. The style loss is defined as

$$\mathcal{L}_{style} = \frac{1}{n|\Phi_{style}|} \sum_{\varphi \in \Phi_{style}} \sum_{i=1}^m SWD(\varphi(\hat{\boldsymbol{y}}_i), \varphi(\hat{\boldsymbol{y}}_{rec})). \tag{6}$$

$\Phi_{style}$ is a set of convolutional layers from the pre-trained $lpips$ network to extract spatial features. Note that the generator learns the style of $\hat{\boldsymbol{y}}_{rec}$ rather than that of $\boldsymbol{y}_{ref}$ to prevent the style of the entity from leaking into the other area. The entity loss is defined as

$$\mathcal{L}_{ent} = \frac{1}{n|\Phi_{ent}|} \sum_{\varphi \in \Phi_{ent}} \sum_{i=1}^m SWD(\varphi(\boldsymbol{m}_i^\uparrow \odot \boldsymbol{y}_i), \varphi(\boldsymbol{m}_{rec}^\uparrow \odot \boldsymbol{y}_{rec})). \tag{7}$$

$\Phi_{ent}$ is also a set of convolutional layers from the pre-trained $lpips$ net, $\boldsymbol{m}^\uparrow$ is upsampled by $\boldsymbol{m}$ to get the same resolution as the final synthesis. Taking the reconstructed entity $\boldsymbol{m}_{rec}^\uparrow \odot \boldsymbol{y}_{rec}$ as the target can be viewed as an implicit augmentation technique to avoid overfitting.

### 4.4   Manifold regularization

The training losses, particularly the style loss, inevitably distort the content of source syntheses and change their relative relationship. To achieve cross-domain correspondence, we use the variant

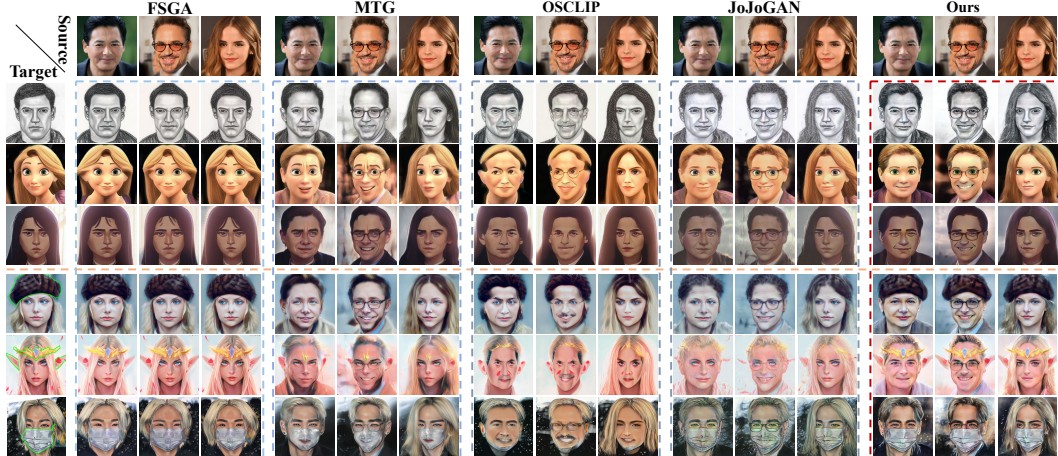

Figure 3: Comparison of our framework with other state-of-the-arts. The source domain contains natural face images of $1024 \times 1024$. Please zoom in for better visual effects.

Laplacian regularization $\mathcal{L}_{VLapR}$ to alleviate the distortion. Since we do not tune the StyleGAN mapping network, the $\mathcal{W}$ space coupled with the source generator $G_s$ does not change, so we define

$$\mathcal{L}_{VLapR} = \int_{\mathcal{W}} \left\| \nabla_{\mathcal{W}^\sharp} \phi(G_t(\boldsymbol{w}^\sharp)) - \nabla_{\mathcal{W}^\sharp} \phi(G_s(\boldsymbol{w}^\sharp)) \right\|^2 dP(\boldsymbol{w}), \tag{8}$$

where $\phi$ is a pre-trained semantic extractor, here we use the pre-trained CLIP image encoder [31] that is sensitive to both style and content variation. $\mathcal{L}_{VLapR}$ can be interpreted from two perspectives. The first is the smoothness of the function: Because the integrand also represents the smoothness of the residual $\phi(G_t(\boldsymbol{w}^\sharp)) - \phi(G_s(\boldsymbol{w}^\sharp))$, minimizing $\mathcal{L}_{VlapR}$ expects syntheses from $G_s$ and $G_t$ across $\mathcal{W}$ have a smooth semantic difference. In the ideal case, the integrand is nearly zero, for various $\boldsymbol{w}$ in a local area, $G_t(\boldsymbol{w}^\sharp)$ and $G_s(\boldsymbol{w}^\sharp)$ have similar differences in terms of style and entity. The second perspective is the preservation of geometric structure: According to the Laplacian regularization theory [3], the integral form in Eq. (8) can be estimated in discrete form;

$$\mathcal{L}_{VLapR} = \sum_{i,j=1}^{n} w_{i,j} \left\| \phi(\boldsymbol{y}_i) - \phi(\boldsymbol{y}_j) + \phi(\boldsymbol{x}_j) - \phi_j(\boldsymbol{x}_i) \right\|^2 \tag{9a}$$

$$= 2tr(\boldsymbol{R}^T \boldsymbol{L} \boldsymbol{R}), \tag{9b}$$

where $\boldsymbol{R} = [\phi(\boldsymbol{y}_1)^T - \phi(\boldsymbol{x}_1)^T; \dots; \phi(\boldsymbol{y}_n)^T - \phi(\boldsymbol{x}_n)^T]$, $w_{i,j} = e^{-\left\| \boldsymbol{w}_i^\sharp - \boldsymbol{w}_j^\sharp \right\| / \sigma}$, $\boldsymbol{L}$ is the Laplacian matrix (see Sec. 3). In Eq. (9a), if $\boldsymbol{w}_i$ and $\boldsymbol{w}_j$ are close in latent space, $w_{i,j}$ will be a large scalar that encourages $\left\| \phi(\boldsymbol{y}_i) - \phi(\boldsymbol{y}_j) \right\|$ and $\left\| \phi(\boldsymbol{x}_i) - \phi(\boldsymbol{x}_j) \right\|$ to be the same. This means that the relative distances of adjacent syntheses before and after adaptation are isometric in the feature space. In practice, the loss can be efficiently computed using Eq. (9b), the derivation is provided in the supplementary materials.

**Relation between $\mathcal{L}_{VlapR}$ and $\mathcal{L}_{CDC}$** Recently the influential work FSGA [29] introduces the cross-domain distance consistency loss $\mathcal{L}_{CDC}$ to achieve cross-domain correspondence. For arbitrary $\boldsymbol{w}_i$ and $\boldsymbol{w}_j$, without loss of generality, FSGA defines the conditional probability $p_{j|i}$ and $q_{j|i}$ in Eq. (10) to form the similarity distributions, then minimizes their Kullback-Leibler divergence Eq. (11) that encourages the syntheses of $G_t$ to have the same similarity property as that of $G_s$.

$$p_{j|i} = \frac{e^{-\left\| \phi(\boldsymbol{x}_j) - \phi(\boldsymbol{x}_i) \right\|^2}}{\Sigma_{j \neq i}^n e^{-\left\| \phi(\boldsymbol{x}_j) - \phi(\boldsymbol{x}_i) \right\|^2}}, \; q_{j|i} = \frac{e^{-\left\| \phi(\boldsymbol{y}_j) - \phi(\boldsymbol{y}_i) \right\|^2}}{\Sigma_{j \neq i}^n e^{-\left\| \phi(\boldsymbol{y}_j) - \phi(\boldsymbol{y}_i) \right\|^2}}. \tag{10}$$

$$\mathcal{L}_{CDC} = \Sigma_{i=1}^n KL(p_{\cdot|i} \| q_{\cdot|i}) = \Sigma_{i=1}^n \Sigma_{j \neq i}^n p_{j|i} \log \frac{p_{j|i}}{q_{j|i}}. \tag{11}$$

In the above equation, $\phi$ denotes the feature map of the corresponding generator. From the perspective of manifold learning, $p_{\cdot|i}$ and $q_{\cdot|i}$ depict the neighborhood structure of $\boldsymbol{x}_i$ and $\boldsymbol{y}_i$ respectively, $\mathcal{L}_{CDC}$ tries to place $\boldsymbol{x}_i$ in a new space so as to optimally preserve neighborhood identity. Actually, we find that this idea is equivalent to the Stochastic Neighbor Embedding [12, 40], which is also a popular

Table 1: Quantitative results. The target images are shown in Fig. 3. US denotes the user preference. NME and IS are not counted on the Mask since the masked face cannot be identified by [7] and [10].

| | Sketch | | | Disney | | | Arcane | | | Hat | | | Zelda | | | Mask |
|---|---|---|---|---|---|---|---|---|---|---|---|---|---|---|---|---|
| | NME↓ | IS↑ | US↑ | NME↓ | IS↑ | US↑ | NME↓ | IS↑ | US↑ | NME↓ | IS↑ | US↑ | NME↓ | IS↑ | US↑ | US↑ |
| FSGA | 0.14 | 0.11 | 0.09 | 0.26 | 0.00 | 0.09 | 0.17 | 0.04 | 0.09 | 0.19 | 0.03 | 0.10 | 0.28 | 0.00 | 0.09 | 0.10 |
| MTG | 0.08 | 0.31 | 0.09 | 0.19 | 0.14 | 0.11 | 0.08 | 0.24 | 0.10 | 0.09 | 0.29 | 0.10 | 0.15 | 0.05 | 0.09 | 0.10 |
| JoJoGAN | 0.09 | 0.24 | 0.13 | 0.11 | 0.17 | 0.11 | 0.10 | 0.17 | 0.11 | 0.08 | 0.21 | 0.10 | 0.09 | 0.09 | 0.11 | 0.10 |
| OSCLIP | 0.11 | 0.20 | 0.09 | 0.11 | 0.19 | 0.11 | 0.10 | 0.19 | 0.09 | 0.13 | 0.23 | 0.10 | 0.21 | 0.13 | 0.10 | 0.10 |
| Ours | **0.08** | **0.40** | **0.60** | **0.09** | **0.26** | **0.58** | **0.07** | **0.25** | **0.61** | **0.08** | **0.36** | **0.60** | **0.09** | **0.25** | **0.61** | **0.60** |
| w/o $\mathcal{L}_{style}$ | 0.07 | 0.45 | - | 0.07 | 0.40 | - | 0.07 | 0.32 | - | 0.08 | 0.40 | - | 0.08 | 0.35 | - | - |
| w/o $\mathcal{L}_{VLapR}$ | 0.09 | 0.35 | - | 0.10 | 0.11 | - | 0.08 | 0.21 | - | 0.10 | 0.29 | - | 0.12 | 0.15 | - | - |

method for preserving the manifold structure. $\mathcal{L}_{CDC}$ has the following disadvantages. Firstly, a large batch is almost inacceptable for a high-resolution GAN to estimate $p_{.|i}$ and $q_{.|i}$. $\mathcal{L}_{VLapR}$ directly constrains the distance and performs well even when the batch is 2. Second, $\mathcal{L}_{CDC}$ is non-convex and hard to optimize, in practice, the loss value is usually around 1e-5, implying that $\mathcal{L}_{CDC}$ does not provide an effective penalty for the differences. Finally, because the $softmax$ form of $p_{j|i}$ and $q_{j|i}$ is scale-invariant about the inputs, $\mathcal{L}_{CDC}$ cannot guarantee the isometric relationship of the source and adapted syntheses like $\mathcal{L}_{VLapR}$, thus it is hard to avoid mode collapse. In the supplementary materials, we show that replacing $\mathcal{L}_{CDC}$ with $\mathcal{L}_{VLapR}$ can greatly remedy the collapse of FSGA.

# 5 Experiments

Our implementation is based on the official code of StyleGAN[1]. If $m_{ref} \neq 0$, the total epoch is 2000. We adopt the Adam optimizer with learning rate $1e-3$, $\beta_1 = 0$, $\beta_2 = 0.999$. The cosine annealing strategy of the learning rate is adopted to reduce the learning rate to $1e-4$ gradually. In Eq. (2), $\lambda_1 = 10$, $\lambda_2 = 0.2$, $\lambda_3 = 2$, $\lambda_4 = 1$, and in Eq. (4) $\lambda_5 = 100$. We use the pre-trained $lpips$ network with $vgg$ architecture [38] containing five convolutional blocks. In Eq. (6) and Eq. (7), $\Phi_{style} = \{vgg_{1:4}, vgg_{1:5}\}$, $\Phi_{ent} = \{vgg_{1:1}, vgg_{1:2}, vgg_{1:3}, vgg_{1:4}\}$, and $m = 1$. In Eq. (9b), $n = 2$ including the samples from reconstruction and internal distribution learning. $\sigma$ for calculating the graph weights is 128. We use the Monte Carlo simulation that randomly samples 256 vectors on the unit sphere to compute the integral in Eq. (5). If $m_{ref} = 0$, we change $\lambda_2$ to 2, $\lambda_4$ to 0.5, and epoch to 1000 for speeding up training. The augmentation of the reference only includes the horizontal flip. More experimental details and results can be found in the supplementary materials.

## 5.1 Comparison with SOTA methods

We take the few-shot adaptation method FSGA [29], one-shot adaptation methods MTG [55], OSCLIP [20], and one-shot face style transfer method JoJoGAN [9] for comparison on the commonly studied face domain. As shown in Fig. 3, since these works cannot deal with the entity yet, for a fair comparison we select three portraits without entity (Sketch, Disney, Arcane) from previous works [34, 9] and three portraits with entities (hat, Zelda decorations, and mask) from AAHQ [23]. Their masks can be obtained via manual annotation, or alternatively by semantic segmentation [54, 24].

**Qualitative results** Fig. 3 shows the qualitative results. The top row images are the source natural faces, and they are inverted by e4e [39] to obtain the latent codes. From the results, we can draw the following conclusions: Firstly, our syntheses are of competitive style as the given references. Though shown in thumbnails, they have more obvious high-frequency details to look sharp and realistic, whereas the results of JoJoGAN are very fuzzy. Secondly, our method achieves the strongest cross-domain correspondence that preserves the content and shape of the source image, which is a great advance compared with other methods without a doubt. It also implies our method can keep the diversity of the source model without content collapse. Thirdly, our method can generate high-quality entities, which are adaptive to the various face shapes and look harmonious with the rest area of the syntheses. Experiments prove that our method has sufficient visual advantages over competitors.

**Quantitative results** Previous works mainly conduct user studies to compare the visual quality. Following them, we survey user preferences for different methods. We first randomly sample 50 latent codes in $\mathcal{W}$ space and feed them into various models. We provide the references, source syntheses,

---

[1] https://github.com/NVlabs/stylegan3

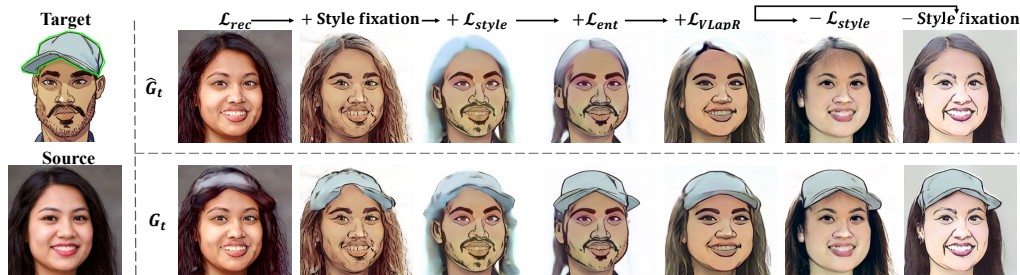

Figure 4: Ablation study. The arrows indicate how the current model has changed from the previous model. The upper and lower syntheses come from $\hat{G}_t$ and $G_t$ respectively.

and adapted syntheses to volunteers. They consider both style and content and subjectively vote for their favorite adapted syntheses. Moreover, to objectively evaluate the cross-domain correspondence, we use the face alignment network [7] to extract facial landmarks and calculate the Normalized Mean Error (NME) of landmarks between source syntheses and adapted syntheses.

$$NME(G_s, G_t) = \frac{1}{n} \sum_{i=1}^{n} \frac{\|lmk(\boldsymbol{x}_i) - lmk(\boldsymbol{y}_i)\|^2}{\sqrt{HW}}, \tag{12}$$

where $lmk$ denotes the 68 points two-dimension landmarks extractor, and $n$ is set to 1000. NME reflects the face shape difference before and after adaptation. We further report the Identity Similarity (IS) predicted by the Arcface [10] to metric the preservation of identity information. The model with lower NME and higher IS shows better cross-domain corresponding property. The quantitative results have been shown in Table 1. Our method undoubtedly obtains the best scores in terms of NME and IS. It performs stably and gets similar NME in different domains. FSGA and OSCLIP obtain relatively worse scores due to overfitting as shown in Fig. 3, their common characteristic is that the GAN discriminator is used for training. MTG performs very unsteadily especially when the target domains have strong semantic style like Disney and Zelda. Compared with our approach, JoJoGAN works well on NME, but poorly on IS. For the user study, we finally collect valid votes from 53 volunteers, and a user averagely spends 4 seconds on one question. Our method obtains about 60% of the votes on each adapted model, whereas other methods evenly acquire the rest of the votes. It proves that our method is much more popular with users.

**Training and inference time**    Our method costs about 12 minutes for $\boldsymbol{m}_{ref} \neq \boldsymbol{0}$ and 3 minutes for $\boldsymbol{m}_{ref} = \boldsymbol{0}$ on NVIDIA RTX 3090. FSGA, MTG, JoJoGAN, OSCLIP cost about 48, 9, 2, 24 minutes respectively. All of them take about 30ms to generate an image.

## 5.2    Ablation study

We start from the basic reconstruction loss $\mathcal{L}_{rec}$ and continuously improve the framework to observe the qualitative changes of the syntheses. As illustrated in Fig. 4, the target exemplar is a hard case depicting a man with a beard and cap, and style fixation can bring a rough but remarkable improvement to the style of synthesis. Although $\mathcal{L}_{style}$ can enhance the style of the face, it deteriorates the hair and introduces noise like the beard to the female face. $\mathcal{L}_{ent}$ plays a critical role in the generation of the entity. Without it, the cap cannot be synthesized. In particular, $\mathcal{L}_{VLapR}$ has a significant suppression of the noise in previous syntheses. Finally, we remove $\mathcal{L}_{style}$ and style fixation respectively, then the syntheses are no longer of the target style. It means that both of them play vital roles in style transfer. We also provide the quantitative ablation in Table 1 to study the impacts of $\mathcal{L}_{style}$ and $\mathcal{L}_{VLapR}$. It shows that removing $\mathcal{L}_{style}$ has a positive effect on NME and IS while removing $\mathcal{L}_{VLapR}$ has an opposite effect. The quantitative results confirm that $\mathcal{L}_{VLapR}$ is necessary for preventing content from being distorted by style loss.

## 5.3    More results

**Adaptation of various domains**    In addition to the success of the face domain, our framework can be applied to other source domains. We take StyleGANs pre-trained on the $512 \times 512$ AFHQ dog, cat data set [8] and the $256 \times 256$ LSUN church dataset [48] as exemplars. As shown in Fig. 5, for each source domain, we provide images with entities, and without entities, the source images are

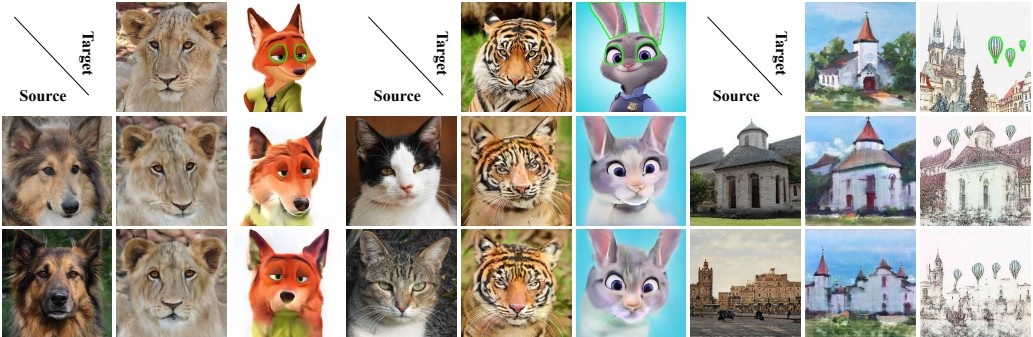

Figure 5: Our framework can transfer various source domains to various target domains.

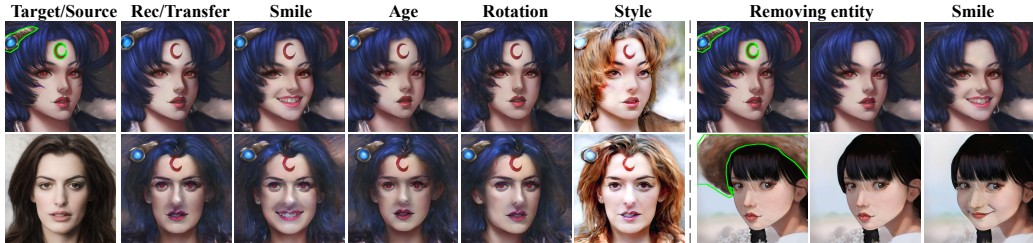

Figure 6: Applying the adapted model to manipulate target image and transferred image, including semantic editing, style transfer and entity removal.

randomly sampled from the source space. For images without entities, the adapted syntheses are realistic and have a strong corresponding relationship in visual. In other cases, our approach also yields satisfactory results. For example, the cartoon eyes of the dogs are in the right place and very similar to that of the reference. It proves that our framework is valid and has a wide universality.

**Image manipulation** Because our adapted model retains the geometric structure of the source generative space, it can efficiently perform semantic editing on the image using the semantic direction found in the source space. As shown in Fig. 6, since our model is capable of accurately reconstructing the target exemplar, we can edit both the exemplar and the synthesis. We select three representative semantic directions [36] to change smile, age, and pose rotation, and the results show that the adapted generator achieves exact control not only on the face but also on the entity. In addition, our model can also change the style by altering the style code, which is hard for previous works. Furthermore, our framework can be applied to remove the entity with $\hat{G}_t$, and the result can be manipulated similarly. We consider these functions will be helpful for artistic creation.

## 6 Conclusion and limitations

In this paper, we construct a new GAN adaptation framework for the generalized one-shot GAN adaptation task. Because our framework relies heavily on learning of internal distributions, it inevitably has some limitations. We think the most important one is that it cannot precisely control the position of the entity, which may lead to failures when the pose changes too much. Another limitation is that the entity cannot be too complex, otherwise, it is difficult to learn by patch distribution. However, these extreme cases require additional consideration. In general, our framework is effective in various scenarios. We believe that the exploration of the novel task and the introduction of manifold regularization are significant for future work.

## Acknowledgements

This paper is supported by the National key research and development program of China (2021YFA1000403), the National Natural Science Foundation of China (Nos. U19B2040), the Strategic Priority Research Program of Chinese Academy of Sciences, Grant No. XDA27000000, and the Fundamental Research Funds for the Central Universities.

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
