# Generalized One-shot Domain Adaptation of Generative Adversarial Networks Supplementary Materials

Zicheng Zhang[1]    Yinglu Liu[2]    Congying Han[1]    Tiande Guo[1]    Ting Yao[2]    Tao Mei[2]

[1]University of Chinese Academy of Sciences    [2]JD AI Research

zhangzicheng19@mails.ucas.ac.cn    liuyinglu1@jd.com    hancy@ucas.ac.cn
tdguo@ucas.ac.cn    tingyao.ustc@gmail.com    tmei@live.com

## Contents

# 1 Effects of the introduction of the entity mask

The introduction of the extra entity mask has huge significance for one-shot adaptation task, to illustrate that we use the same references with and without masks to adapt the GANs. The results in Fig. 1(a) prove the mask helps to clearly define the target domain. If there is no mask, the synthesis only obtains the exemplary style, otherwise both the entity and style. Fig. 1(b) shows that, without a mask, the hair is polluted by the color and texture of the Christmas hat, thus a mask can prevent the style of the entity from negatively impacting other areas. In Fig. 1(c), the mask allows the model to pay more attention to the interesting objects, note that the eyes on the right are closer to the reference than the eye on the left.

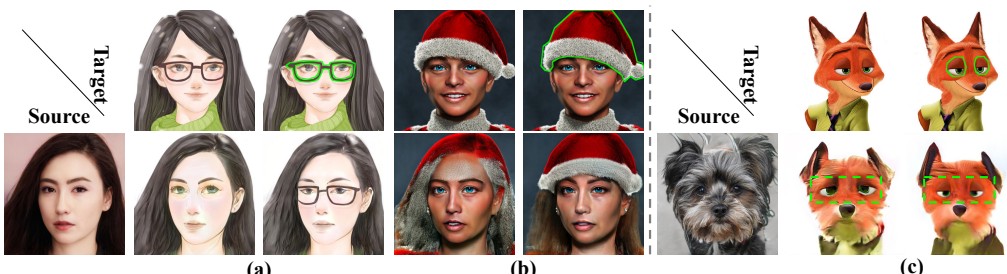

Figure 1: Adapting the GANs with and without entity masks can obtain different results.

# 2 Comparison of different distribution matching losses

Both style transfer and entity generation can be interpreted by learning the internal distribution of the example, thus we compare the SWD with the commonly-used Gram loss [5], moment matching loss [10, 7] to demonstrate the superiority of SWD. Although patch GAN loss [6] is an alternative in theory and the most prevalent way for image translation, the GAN framework is too bulk (about 50 minutes training time, large GPU memory occupation), and easily causes serious over-fitting for the single target image. Please refer to the right image in Fig. 3 to see the results of using patch GAN loss.

In theory, SWD can completely capture the distribution. For two distributions $p$ and $q$, $SWD(p, q) = 0 \Leftrightarrow p = q$ [14, 8]. However, as proved in [10], Gram loss vanishes just means $p$ and $q$ have same expectation (*i.e.*, the first center moment). And the moment matching loss [10, 7] vanishes just means $p$ and $q$ have the same high order center moments. Hence, SWD, Gram loss, and moment matching loss are theoretically practicable for Style adaptation. It is well known that a successful generative model needs learn the exact distribution, thus only SWD can be used for entity adaptation.

We conduct the experiment to validate the theoretical analysis. We adopt Gram loss [5], moment matching loss [10], and SWD to $\mathcal{L}_{style}$ and $\mathcal{L}_{ent}$ respectively. They will run with the same vgg features as stated in the main paper. After careful adjustment of weights, for Gram loss we set the weight of $\mathcal{L}_{style}$ as $2e-6$, the weight of $\mathcal{L}_{ent}$ as $2e-5$. For moment loss we set the weight of $\mathcal{L}_{style}$ as $2e-3$, the weight of $\mathcal{L}_{ent}$ as $2e-2$. The results are shown in Fig. 2. These three losses can get similar visual effects for style adaptation, but behave very differently for entity adaptation. Obviously, Gram loss gets the worst results. Hence most pixels in entity masks are zero, and the mean value of the target entity image is close to zero, causing the synthesized entities to be grey. The moment matching loss can precisely transfer the red color of the hats than Gram loss, but it still cannot transfer the glass completely. Finally, SWD can achieve the best results. The experiment proves that SWD is more suited to our task than other distribution matching losses.

# 3 Laplacian regularization

## 3.1 PyTorch-style pseudo code

The PyTorch-style pseudo code is shown in Algorithm 1, note that a normalization process is needed to eliminate the impact of batch and channel number at last.

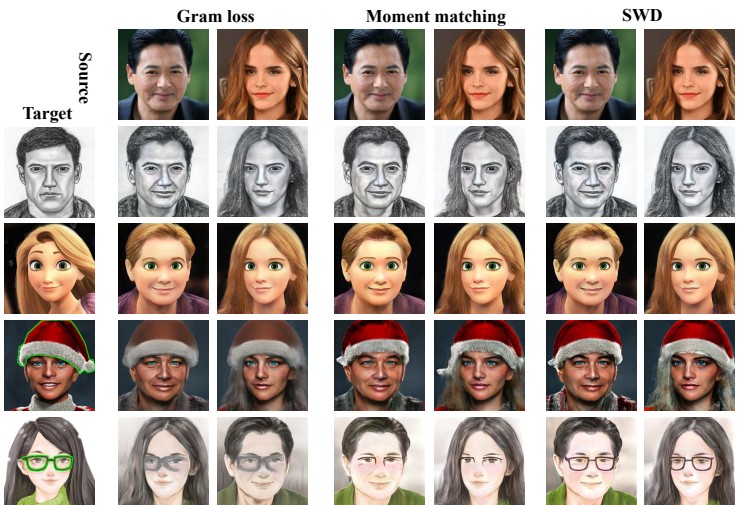

Figure 2: Comparisons of different internal distribution losses.

---

**Algorithm 1:** PyTorch-style pseudocode for $\mathcal{L}_{VLapR}$

---

```
# encoder - CLIP (or other) image encoder
# z [B,C] or [B,d,C] - latent codes
# source_sample [B,C,H,W] - syntheses from source generator
# target_sample [B,C,H,W] - syntheses from target generator
# t - scalar
# compute Laplacian matrix
z = z.flatten(1)
W = torch.exp(-(z.unsqueeze(1)-z.unsqueeze(0)).norm(dim=-1, p=2)/t)
D = torch.diag(torch.sum(W, dim=-1))
L = D-W
# compute loss
R = encoder(target_sample)-encoder(source_sample)
loss = torch.trace(R.permute(1, 0)@L@R)
# eliminate the influence of B and C
B,C = R.shape
loss /= ((B**2-B)*C)
```

---

## 3.2 Derivation

In the main paper we have proposed the

$$\mathcal{L}_{VLapR} = \int_{\mathcal{W}} \left\| \nabla_{\mathcal{W}^\sharp} \phi(G_t(\boldsymbol{w}^\sharp)) - \nabla_{\mathcal{W}^\sharp} \phi(G_s(\boldsymbol{w}^\sharp)) \right\|^2 dP(\boldsymbol{w}), \tag{1}$$

which can be estimated by

$$\mathcal{L}_{VLapR} = \sum_{i,j=1}^{n} w_{i,j} \left\| \phi(\boldsymbol{y}_i) - \phi(\boldsymbol{y}_j) + \phi(\boldsymbol{x}_j) - \phi_j(\boldsymbol{x}_i) \right\|^2 \tag{2a}$$

$$= 2tr(\boldsymbol{R}^T \boldsymbol{L} \boldsymbol{R}). \tag{2b}$$

Here we make a brief description of the relation of the above formulas. Since the derivation only relies on some classical conclusions, we do not present them here but provide the necessary references for readers to consult.

**From Eq.** (1) **to Eq.** (2a)  ① If we define $f(\boldsymbol{w}) = \phi(G_t(\boldsymbol{w})) - \phi(G_s(\boldsymbol{w}))$, since the linearity of derivative is true on Riemannian manifold (Theorem 2.1 in [11]), the Eq. (1) is equivalent to

$$\int_{\mathcal{W}} \left\| \nabla_{\mathcal{W}^\sharp} f(\boldsymbol{w}^\sharp) \right\|^2 dP(\boldsymbol{w}). \tag{3}$$

$P$ is the probability measure of $\boldsymbol{w}$ defined in $\mathcal{W}$. ② Recalling the classic Laplacian regularization $\int_{\mathcal{M}_s} \|\nabla_{\mathcal{M}_s} \boldsymbol{y}\|^2 dP(\boldsymbol{x}) = \int_{\mathcal{M}_s} \boldsymbol{y} \Delta_{\mathcal{M}_s} \boldsymbol{y} dP(\boldsymbol{x})$ (see Preliminaries in the main paper, here we abuse the symbol $P$ to denote the corresponding probability measure). The latter one can be estimated by $\sum_{i,j} w_{i,j} \|\boldsymbol{y}_i - \boldsymbol{y}_j\|^2$ because the Laplacian graph operator and Laplacian operate has a point-wise convergence relation (Theorem 3.1 in [2]). Just replacing $\boldsymbol{y}$ with $f(\boldsymbol{w}^\sharp)$, we can immediately obtain the approximation between Eq. (2a) with

$$\int_{\mathcal{W}^\sharp} \left\|\nabla_{\mathcal{W}^\sharp} f(\boldsymbol{w}^\sharp)\right\|^2 d\mu(\boldsymbol{w}^\sharp), \tag{4}$$

where $\mu$ is the probability measure of $\boldsymbol{w}^\sharp$. ③ Note that $\mu$ is the push-forward measure of $P$, according to the change-of-variables formula (Theorem 1.6.12 in [1]), Eq. (3) is equivalent to (4). Thus we have obtained the approximation between Eq. (1) and Eq. (2a)

**From Eq. (2a) to Eq. (2b)** Let $\boldsymbol{r}_i = \phi(\boldsymbol{y}_i) - \phi(\boldsymbol{x}_i)$, then

$$tr(\boldsymbol{R}^T \boldsymbol{L} \boldsymbol{R}) = tr(\boldsymbol{R}^T \boldsymbol{D} \boldsymbol{R}) - tr(\boldsymbol{R}^T \boldsymbol{W} \boldsymbol{R}) \tag{5}$$

$$= tr(\sqrt{\boldsymbol{D}}\boldsymbol{R}(\sqrt{\boldsymbol{D}}\boldsymbol{R})^T) - tr(\boldsymbol{W}\boldsymbol{R}\boldsymbol{R}^T) \tag{6}$$

$$= \sum_{i=1}^n d_{i,i} \boldsymbol{r}_i^T \boldsymbol{r}_i - \sum_{i=1}^n \left(\sum_{j=1}^n w_{i,j} \boldsymbol{r}_j^T\right) \boldsymbol{r_i} \tag{7}$$

$$= \sum_{i=1}^n \sum_{j=1}^n w_{i,j} \boldsymbol{r}_i^T \boldsymbol{r}_i - \sum_{i=1}^n \sum_{j=1}^n w_{i,j} \boldsymbol{r}_j^T \boldsymbol{r_i} \tag{8}$$

$$= \frac{1}{2}\sum_{i=1}^n \sum_{j=1}^n w_{i,j} \boldsymbol{r}_i^T \boldsymbol{r}_i - \sum_{i=1}^n \sum_{j=1}^n w_{i,j} \boldsymbol{r}_j^T \boldsymbol{r_i} + \frac{1}{2}\sum_{i=1}^n \sum_{j=1}^n w_{i,j} \boldsymbol{r}_j^T \boldsymbol{r}_j \tag{9}$$

$$= \frac{1}{2}\sum_{i=1}^n \sum_{j=1}^n w_{i,j} \|\boldsymbol{r}_i - \boldsymbol{r}_j\|^2 \tag{10}$$

## 3.3 Replacing $\mathcal{L}_{CDC}$ with $\mathcal{L}_{VLapR}$

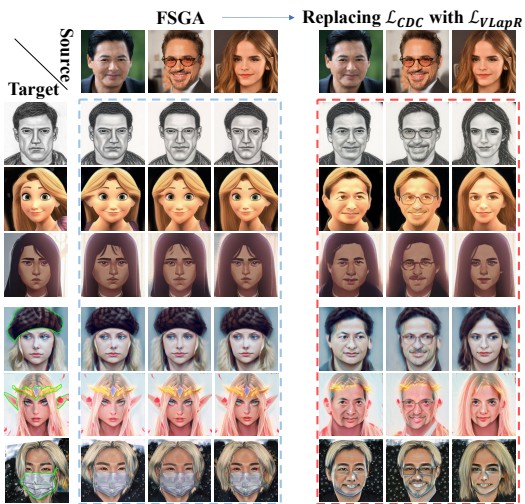

Figure 3: Replacing $\mathcal{L}_{CDC}$ with $\mathcal{L}_{VLapR}$.

We have shown in the article that both $\mathcal{L}_{CDC}$ and $\mathcal{L}_{VLapR}$ can be viewed as the regularization of the manifold in nature. Here we replace $\mathcal{L}_{CDC}$ with $\mathcal{L}_{VLapR}$ to prove that $\mathcal{L}_{VLapR}$ has obvious advantages over $\mathcal{L}_{CDC}$ in maintaining cross-domain correspondence since it is of isometric property in semantic space. We set the weight of $\mathcal{L}_{VLapR}$ to 100, leaving all other configurations of FSGA

[13] unchanged. It can be seen from the Fig. 3 that $\mathcal{L}_{VLapR}$ significantly remedies the collapse of FSGA.

# 4 Auxiliary network

Our auxiliary network includes a UNet architecture ($unet$) [15] shown in Fig. 4. It receives the feature $\boldsymbol{f}_{in}$ from StyleGAN and outputs feature $\boldsymbol{f}_{ent}$ and mask $\boldsymbol{m}$ of entity. Then they will be processed by the Eq. (1) of the main paper. Note that, the predicted mask $\boldsymbol{m}$ is of $256 \times 256$ resolution rather than $32 \times 32$ to label the entity more finely. We do not emphasize it in the main paper to highlight our core idea.

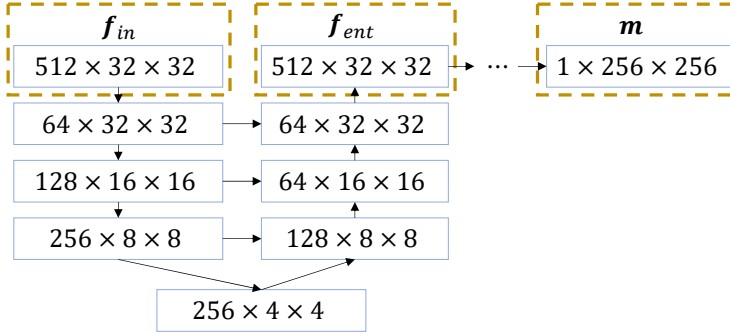

Figure 4: UNet architecture in the auxiliary network. It receives the $\boldsymbol{f}_{in}$ from StyleGAN and outputs the $\boldsymbol{f}_{ent}$ and $\boldsymbol{m}$.

# 5 Failure cases

As we mentioned in the conclusion, our method could be failed for two main cases. As shown in Fig. 5), the first case is the synthesized entities are distorted too much when the internal distribution cannot provide an accurate guide to the shape of the entity. For example, the second row shows the bandages that are of regular shape can be synthesized well, however the synthesized irregular moons or stars on the hair are of poor quality. The second case is the synthesized entities maybe not in the proper place when the heads rotate too much. We believe this is because rotated head images, which are extreme samples, occupy a small percentage of all generated samples so the stochastic optimization algorithm is difficult to optimize for these samples.

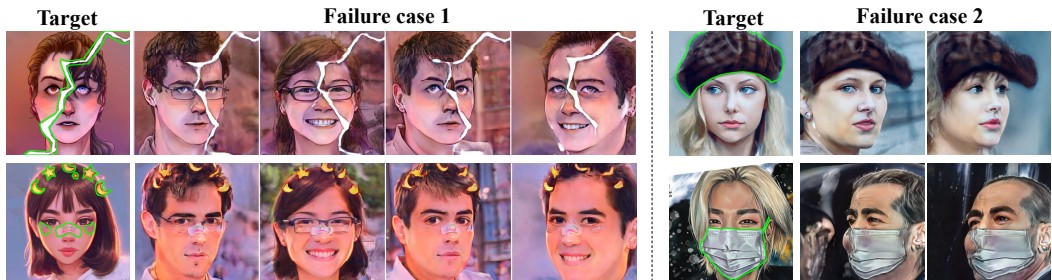

Figure 5: Some failure cases.

# 6 Mask-guided transfer

In our paper, we focus on adapting the GANs to the new domain, thus the framework needs to predict the entity mask of each latent, which is not always accurate, *e.g.*, the right image in Fig. 5. In practice, we may want to only transfer the style and entity for our desired images, or provide a

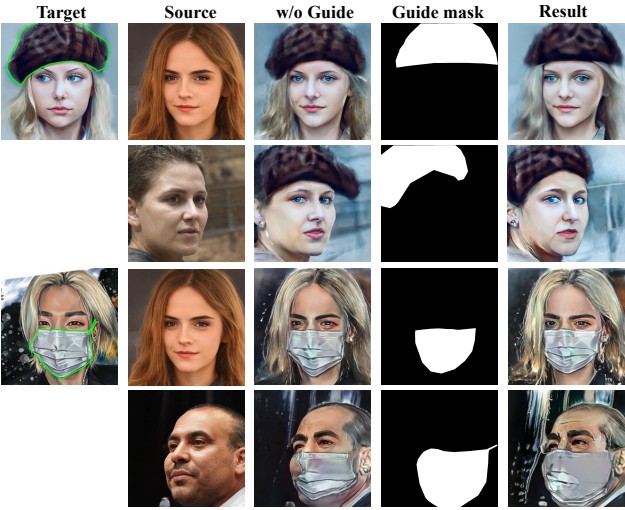

Figure 6: Mask-guided transfer. The third column images are synthesized by our adapted model in the main paper. The user can control the location of the entity by providing the guide mask.

manually annotated mask to improve the location of the entity. This function can be realized by slightly modifying our framework. If the source (content) image corresponds to latent $\boldsymbol{w}_0$ and there exists a mask $\boldsymbol{m}_{user}$ provided by user, we just feed $\boldsymbol{w}_0$ into the framework rather than random latent codes, and constrain the predicted mask $\boldsymbol{m}_0$ with $\|\boldsymbol{m}_0 - \boldsymbol{m}_{user}\|^2$. As shown in Fig. 6, the location entities can be controlled by the guide mask. This would be helpful for artistic creation in practice.

# 7 Broader impact

Our work has the following potential positive impact on society. First, it is resource-efficient and just needs only a single GPU and a small amount of power. Secondly, the generated data avoids manual data collection and privacy security issues. Like other synthesis techniques, our work may have a negative consequence, the generated data may cause fraud in some scenarios, which needs to be solved by developing the DeepFake detection.

# 8 Hyperparameter selection

## 8.1 Style fixation

In the main paper, we have introduced the style fixation,

$$\boldsymbol{w}^{\sharp} = diag(\boldsymbol{\alpha})\boldsymbol{w} + diag(\mathbf{1} - \boldsymbol{\alpha})\boldsymbol{w}_{ref}, \ \alpha_i = \mathbb{1}_{i<=l}(i), \ i = 1, \ldots, 18. \tag{11}$$

In the above equation, the hyper-parameter $l$ is very important to determine how well the content of $\boldsymbol{w}$ is merged with the style of $\boldsymbol{w}_{ref}$. For achieving cross-domain correspondence, the style fixation should ensure that $G_s(\boldsymbol{w})$ and $G_s(\boldsymbol{w}^{\sharp})$ have similar content. We randomly sample two latent codes as $\boldsymbol{w}$ and $\boldsymbol{w}_{ref}$, then compute the similarity between $G_s(\boldsymbol{w})$ and $G_s(\boldsymbol{w}^{\sharp})$ by the pre-trained Arcface model [4]. The process is repeated 10000 times to estimate

$$\mathbf{E}_{\boldsymbol{w} \sim P(\boldsymbol{w}),l} Arcface(G_s(\boldsymbol{w}), G_s(\boldsymbol{w}^{\sharp})), l \in \{0, \ldots, 18\}. \tag{12}$$

The results have been illustrated in Fig. 7, we find setting $l = 8$ obtains $65\%$ average similarity, which is acceptable to say that $G_s(\boldsymbol{w})$ and $G_s(\boldsymbol{w}^{\sharp})$ are the same person. Setting $l = 9$ will make the style of $G_s(\boldsymbol{w}^{\sharp}))$ and reference have obvious visual difference. That means the first eight blocks encode the most content information and the least style information.

## 8.2 Style loss and entity loss

Determining the optimal setting of style loss is very difficult, since a judgment of how good a stylization is depended on aesthetic preferences and remains subjective. By altering the weight of

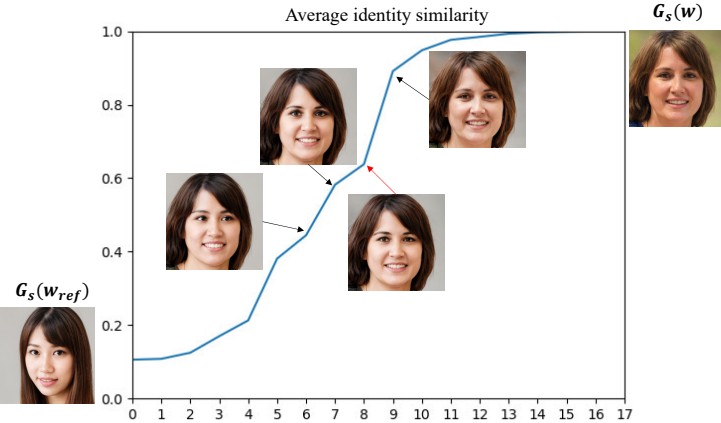

Figure 7: Average identity similarity of syntheses before and after style fixation. The X-axis denotes the change of $l$ in Eq. (11) from 0 to 18. Both $w$ and $w_{ref}$ are randomly sampled from the latent space. The red arrow means the case when $l = 8$.

style loss, our model can generate different syntheses. In our paper, we fix all the weights and the results are very competitive compared with previous methods. In this part, we show some qualitative results to help readers to determine their desired results.

As shown in Fig. 8, applying the features of different layers in the lpips vgg to style loss can make different effects. High-level features are stronger than shallow ones in terms of the representation of style attributes. Choosing which layer to use is very difficult in practice, and a common strategy is to use all of them. However, we find that removing the shallow layers does not influence the effects of stylization, but can save about a quarter of time costs. Hence we use the fourth and fifth layers' features in our paper.

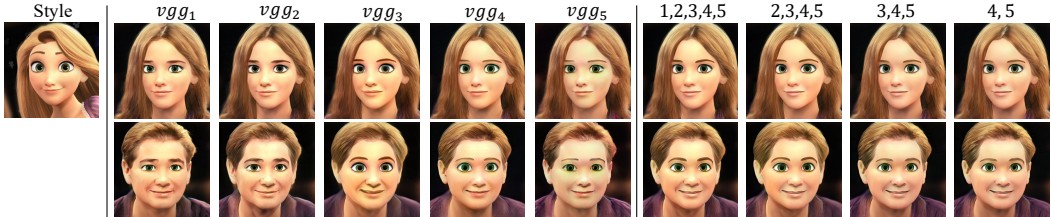

Figure 8: Applying features of different layers of lpips vgg networks to style loss. For the eighth column images, the number $1, 2, 3, 4, 5$ represents it uses the features from $vgg_1$ to $vgg_5$, and so are the others.

For the weight $\lambda_2$ of style loss, Fig. 9 shows that with the increase of $\lambda_2$, the results will be of a more fun style like the target. If we want to keep the original expression and identity information as much as possible, it is best to set this parameter below 2. In the main paper, we just set $\lambda_2 = 2$ if there is no entity, otherwise set $\lambda_2 = 0.2$ to prevent the case that the style loss is too large to influence the entity learning.

For entity loss, a small weight will make the learning of entity slow, and a large weight (above 20 in experience) will make the gradient unstable. It is relatively tractable since entity loss does not distort the content of images like style loss, and we find setting its weight to 2 has a good performance in most cases.

### 8.3 Variant Laplacian regularization

In our experiments, a moderate weight $\lambda_4$ of $\mathcal{L}_{VLapR}$ always has a positive impact on training. A large weight will make the syntheses less distorted by the stylization. As shown in Fig. 10, when the weight is zero, the bears of the target image will pollute the female faces. With the increase of

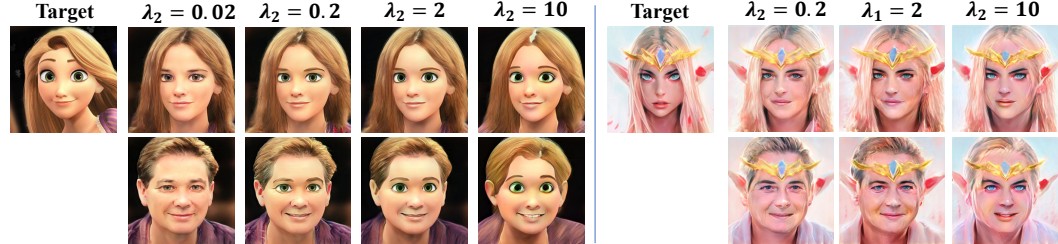

Figure 9: The effect of various style loss weights.

$\lambda_4$, the female faces are more clear in vision. In our paper, if there is only style loss we set it to 0.5, otherwise 1 if there exists an extra loss. However, choosing the proper value requires a trade-off between stylization and content preservation, and it would be better for tuning the weight according to practical usage. Some quantitative results of $\mathcal{L}_{VLapR}$ weight are shown in Table 2.

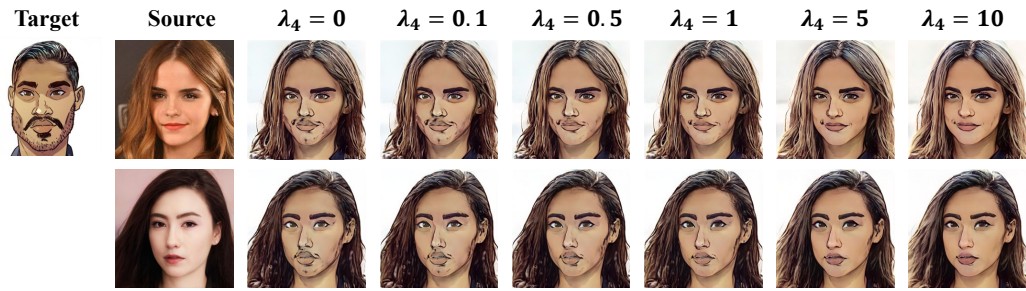

Figure 10: The effect of regularization weights on the results.

## 8.4 Can the auxiliary network be placed elsewhere?

In the main paper, the auxiliary network ($aux$) is placed after the fourth convolutional block to receive the $32 \times 32$ feature map containing the most shape information and least style information. By experiments, we find that placing $aux$ after the fifth convolutional block to receive $64 \times 64$ feature map is an alternative, and anywhere else would be unacceptable. As shown in Fig. 11, when placing it after the third block, the $16 \times 16$ feature map is too small to provide the exact shape information, and the predicted feature map cannot also exactly represent the entity, which leads to the poor quality of the entities. When placing $aux$ after the latter blocks like the sixth block outputting the $128 \times 128$ feature map, the predicted feature map will preferentially lead to the changes in hair or facial style rather than the emergence of new entities. Moreover, predicting the larger feature map is more difficult and requires more computing costs. Hence, we place $aux$ after the fourth block in our paper.

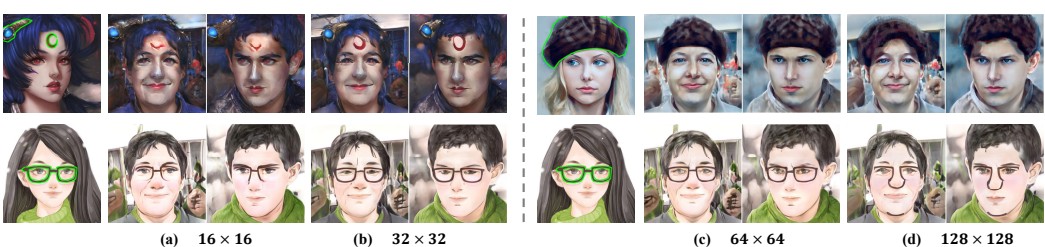

Figure 11: Altering the placement of the auxiliary network to receive and predict the feature maps of different sizes.

Table 1: Average quantitative results of different methods on various domains. The value represents $mean_{std}^{ci}$, and $mean \pm ci$ denotes the confidence interval at 95% confidence level.

| Metric | MTG | OSCLIP | JoJoGAN | Ours |
|---|---|---|---|---|
| NME↓ | $0.12_{0.03}^{0.01}$ | $0.17_{0.10}^{0.04}$ | $0.12_{0.10}^{0.03}$ | $0.10_{0.11}^{0.04}$ |
| ID↑ | $0.16_{0.06}^{0.02}$ | $0.19_{0.03}^{0.01}$ | $0.17_{0.04}^{0.01}$ | $0.27_{0.05}^{0.02}$ |

Table 2: Average quantitative results of different $\mathcal{L}_{VLapR}$ weights on various domains. The value represents $mean_{std}^{ci}$, and $mean \pm ci$ denotes the confidence interval at 95% confidence level.

| Metric | $\lambda_4 = 0$ | $\lambda_4 = 0.5$ | $\lambda_4 = 2$ | $\lambda_4 = 10$ |
|---|---|---|---|---|
| NME↓ | $0.11_{0.10}^{0.03}$ | $0.10_{0.11}^{0.04}$ | $0.10_{0.06}^{0.02}$ | $0.09_{0.06}^{0.02}$ |
| ID↑ | $0.23_{0.05}^{0.02}$ | $0.27_{0.05}^{0.02}$ | $0.30_{0.05}^{0.02}$ | $0.34_{0.04}^{0.02}$ |

# 9 More quantitative results

In the main paper, we follow previous works to report the evaluation of each adapted model. Since the average evaluation on a batch images is more robust, we report the average results on 50 target images collected from [3, 16, 12], where 25 images are with entities and the others are without entities. For each model we generate 1000 samples to calculate the metric as we described in the papper. We evaluate MTG [16], JoJoGAN [3], Oneshot-CLIP [9], and our model on the style adaptation task for the fair comparison. We do not evaluate FSGA, since in the paper we have been shown that FSGA performed serious over-fitting. The results are shown in Table 1. We can conclude that our model has sufficient advantages compared with other methods. It can better preserve the content of source images and has better visual effects (see Sec. 10 for some random samples). The results in Table 2 also proves that the proposed $\mathcal{L}_{VLapR}$ is effective to mitigate the distortion of stylization (see Fig. 10 for qualitative examples.)

# 10 More qualitative results

In this section we present more randomly sampled qualitative results of MTG [16], OSCLIP [9], JoJoGAN [3] and our method. Please refer to Fig. 12 to Fig. 37.

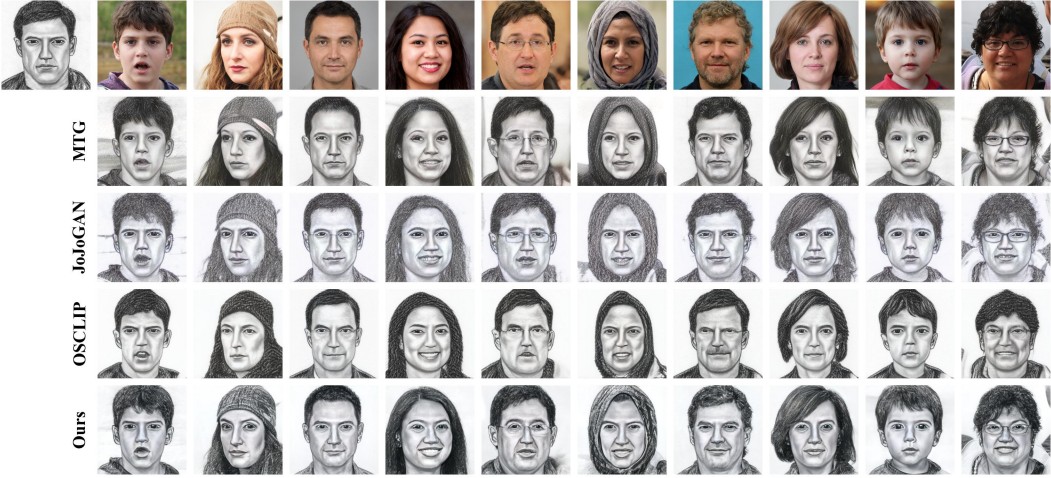

Figure 12: In the first row, the leftmost image is the reference, and the rest of the images are randomly generated by the source generator. The following rows show the adapted results of different methods.

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

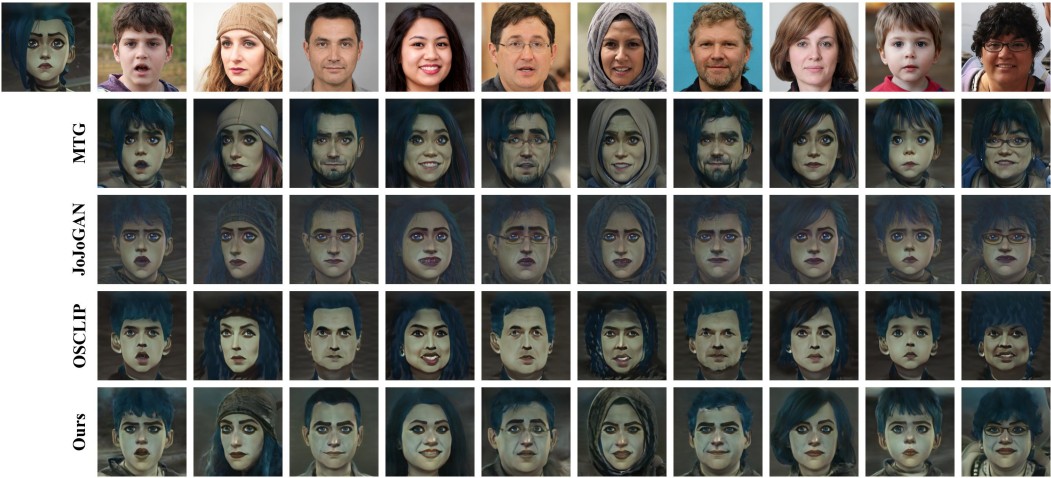

Figure 19: In the first row, the leftmost image is the reference, and the rest of the images are randomly generated by the source generator. The following rows show the adapted results of different methods.

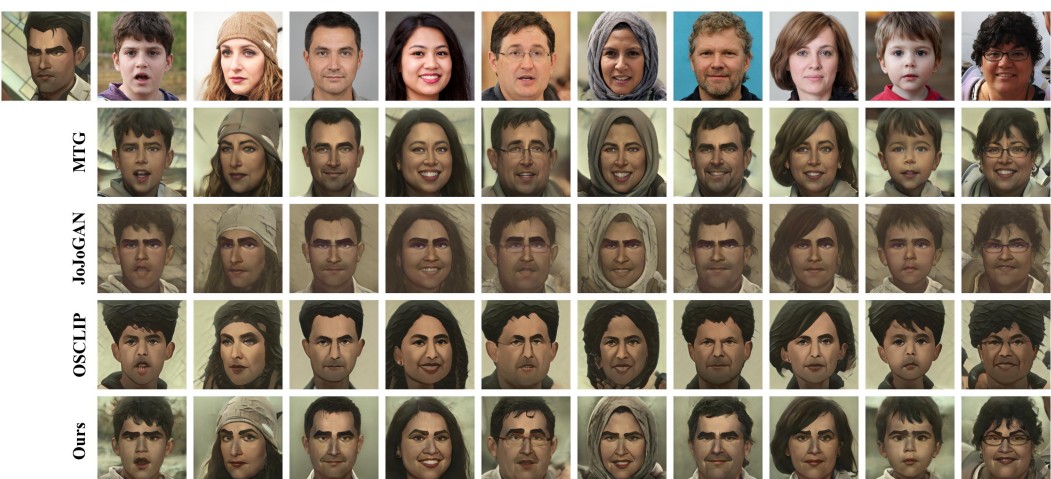

Figure 20: In the first row, the leftmost image is the reference, and the rest of the images are randomly generated by the source generator. The following rows show the adapted results of different methods.

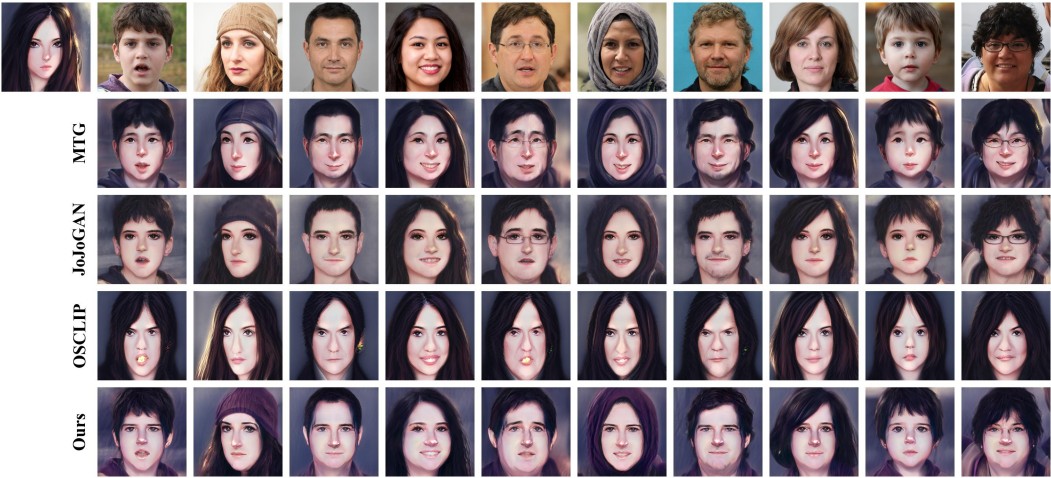

Figure 21: In the first row, the leftmost image is the reference, and the rest of the images are randomly generated by the source generator. The following rows show the adapted results of different methods.

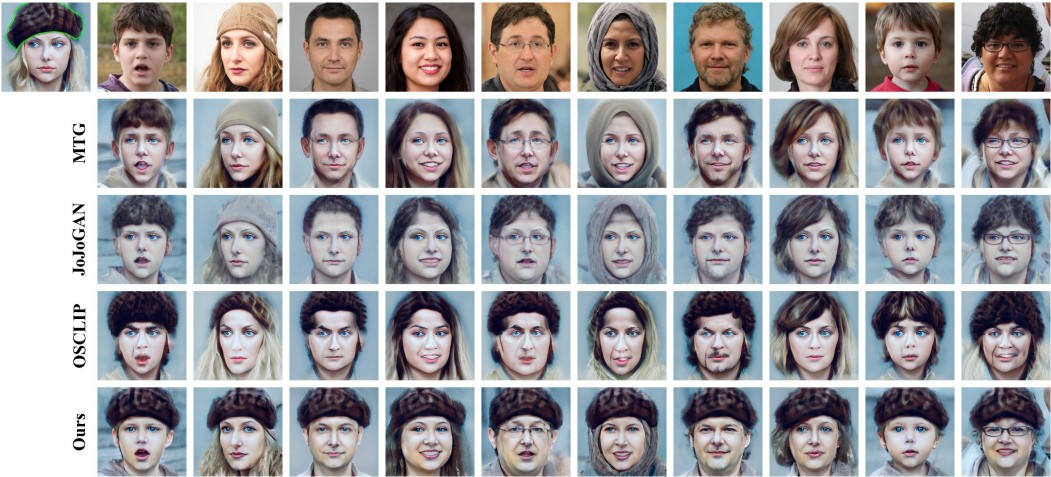

Figure 22: In the first row, the leftmost image is the reference, and the rest of the images are randomly generated by the source generator. The following rows show the adapted results of different methods.

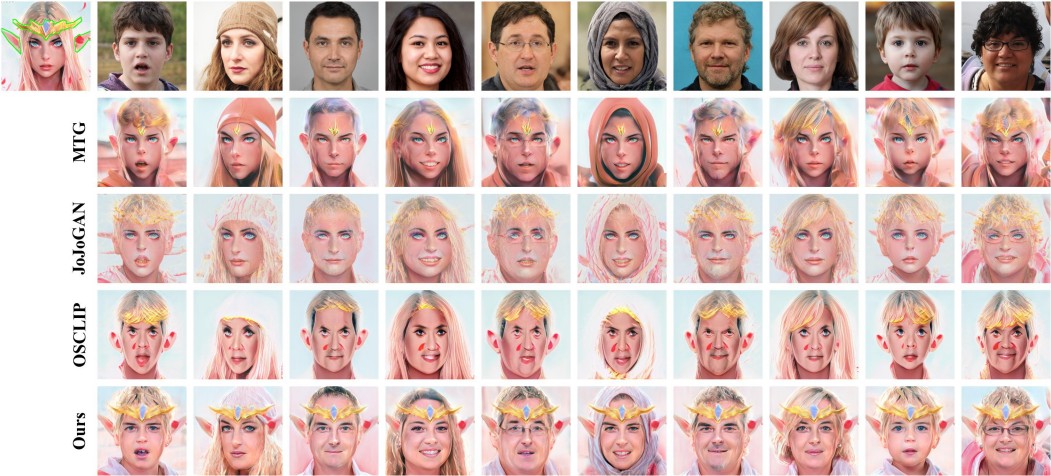

Figure 23: In the first row, the leftmost image is the reference, and the rest of the images are randomly generated by the source generator. The following rows show the adapted results of different methods.

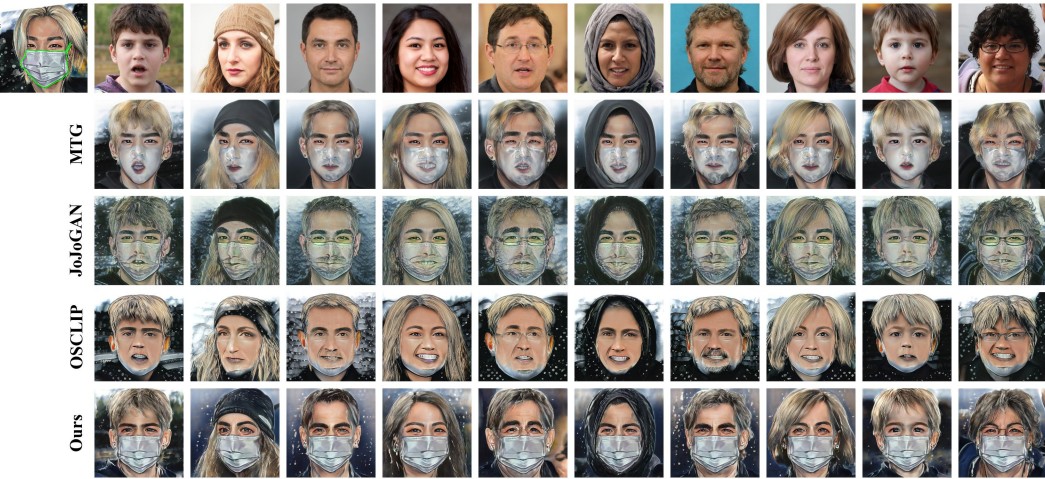

Figure 24: In the first row, the leftmost image is the reference, and the rest of the images are randomly generated by the source generator. The following rows show the adapted results of different methods.

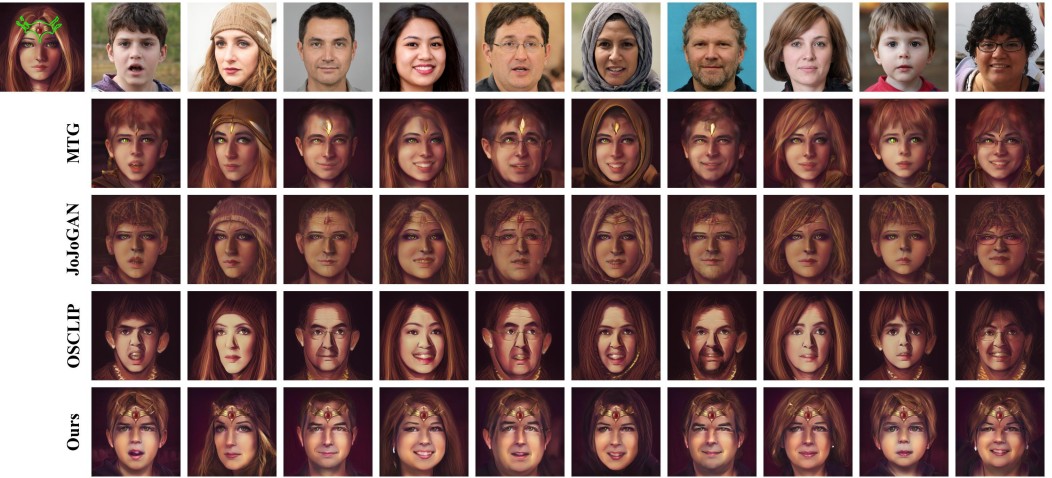

Figure 25: In the first row, the leftmost image is the reference, and the rest of the images are randomly generated by the source generator. The following rows show the adapted results of different methods.

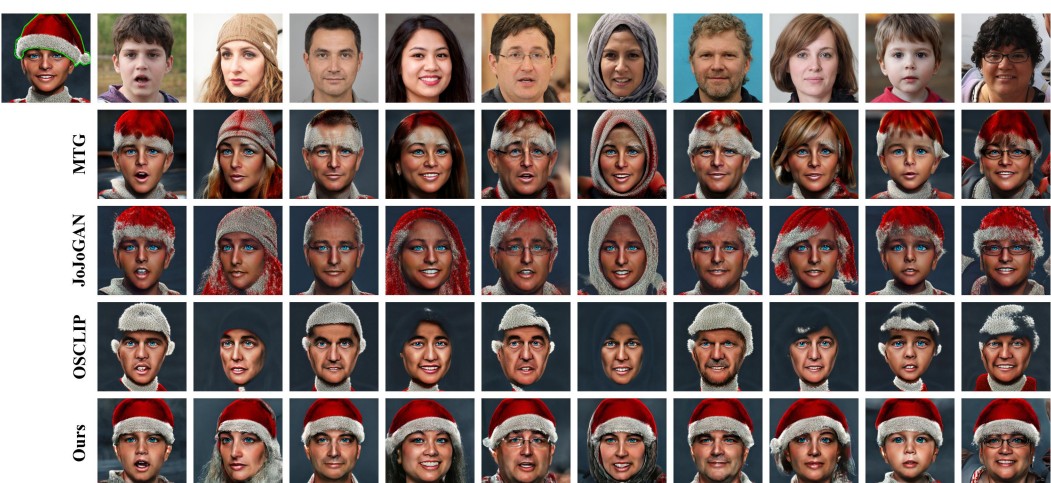

Figure 26: In the first row, the leftmost image is the reference, and the rest of the images are randomly generated by the source generator. The following rows show the adapted results of different methods.

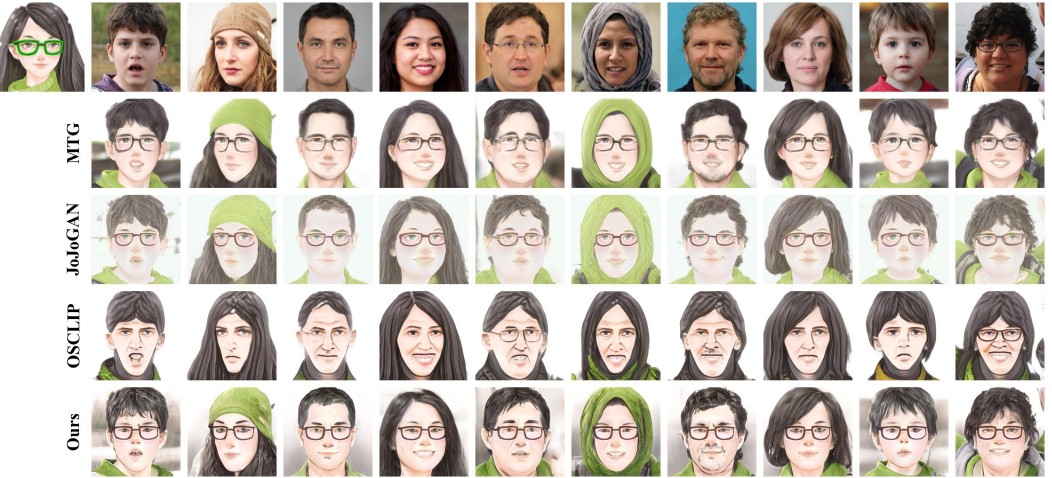

Figure 27: In the first row, the leftmost image is the reference, and the rest of the images are randomly generated by the source generator. The following rows show the adapted results of different methods.

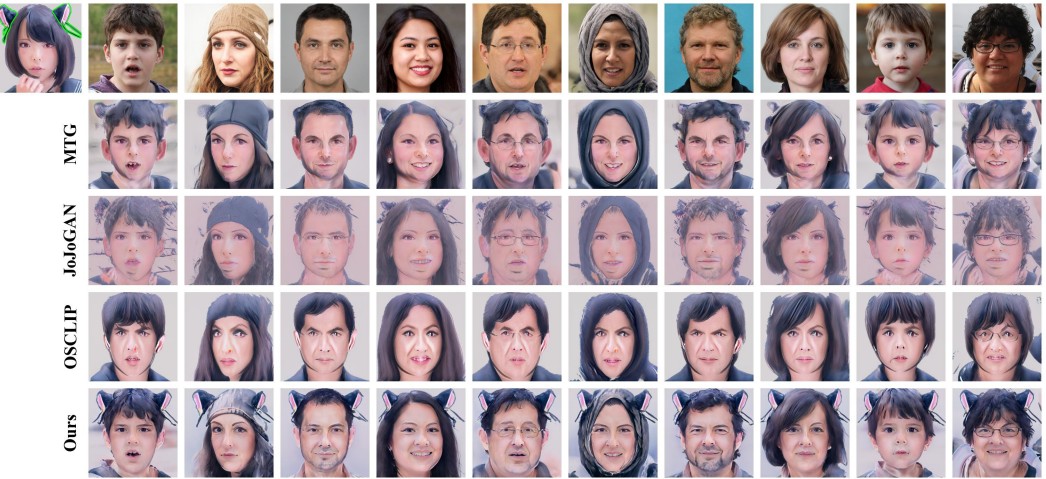

Figure 28: In the first row, the leftmost image is the reference, and the rest of the images are randomly generated by the source generator. The following rows show the adapted results of different methods.

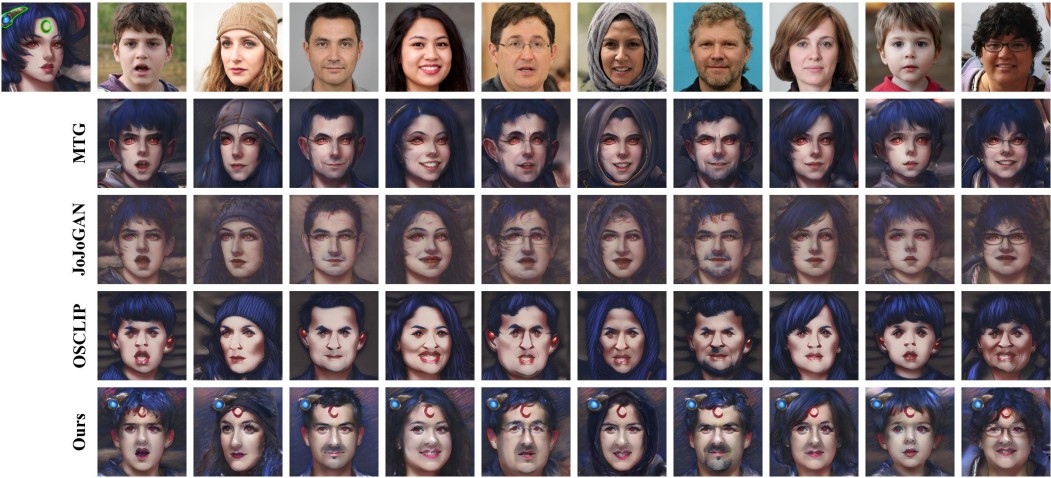

Figure 29: In the first row, the leftmost image is the reference, and the rest of the images are randomly generated by the source generator. The following rows show the adapted results of different methods.

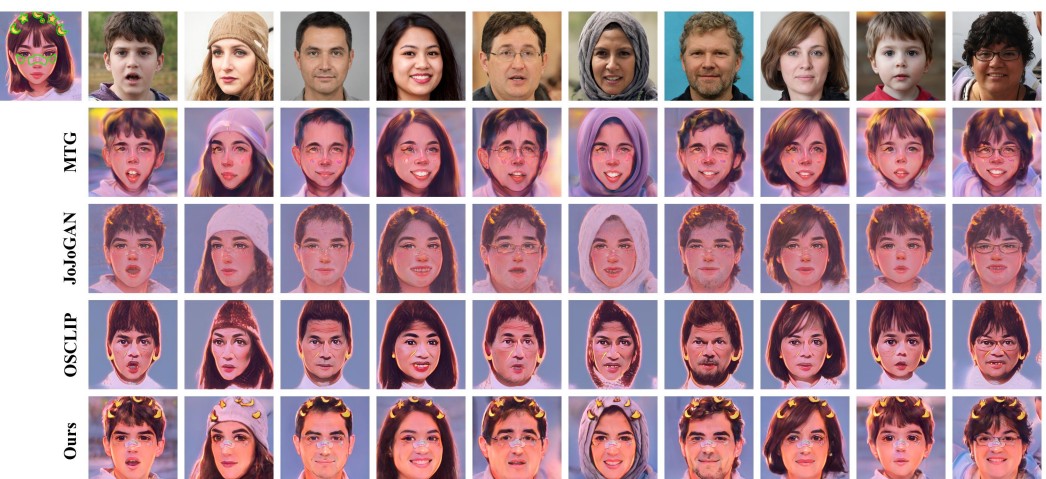

Figure 30: In the first row, the leftmost image is the reference, and the rest of the images are randomly generated by the source generator. The following rows show the adapted results of different methods.

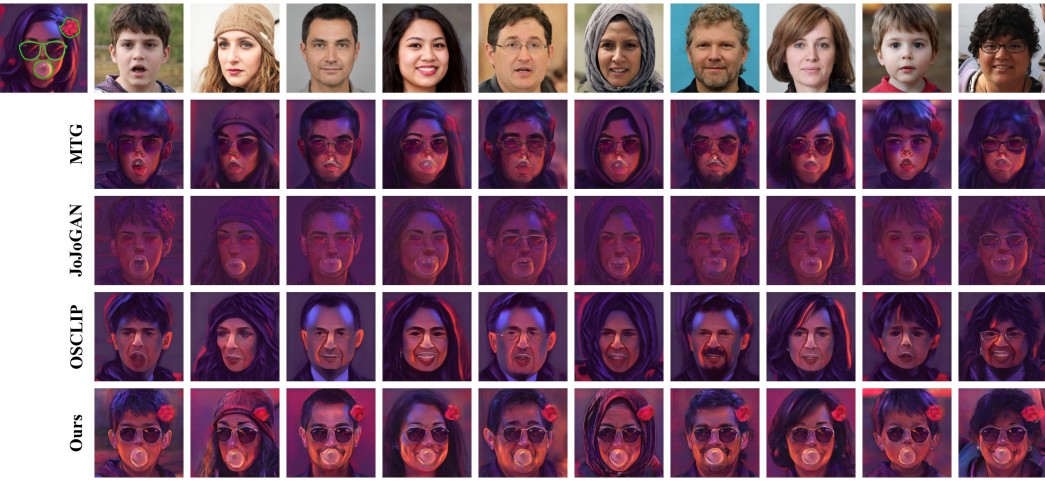

Figure 31: In the first row, the leftmost image is the reference, and the rest of the images are randomly generated by the source generator. The following rows show the adapted results of different methods.

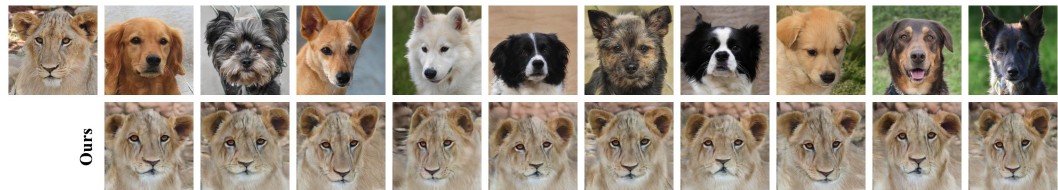

Figure 32: In the first row, the leftmost image is the reference, and the rest of the images are randomly generated by the source generator. The next row shows the adapted results of our method.

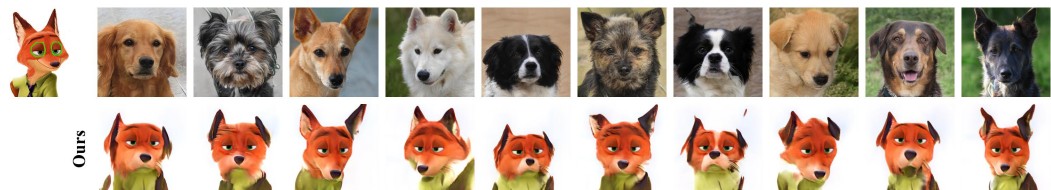

Figure 33: In the first row, the leftmost image is the reference, and the rest of the images are randomly generated by the source generator. The next row shows the adapted results of our method.

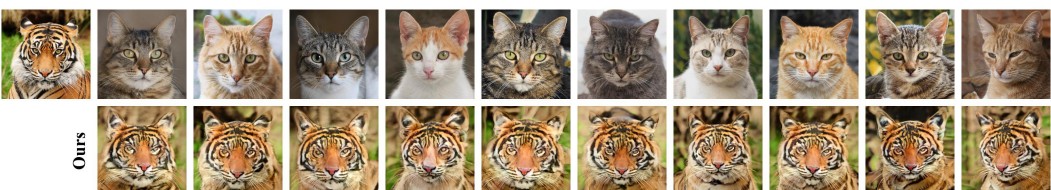

Figure 34: In the first row, the leftmost image is the reference, and the rest of the images are randomly generated by the source generator. The next row shows the adapted results of our method.

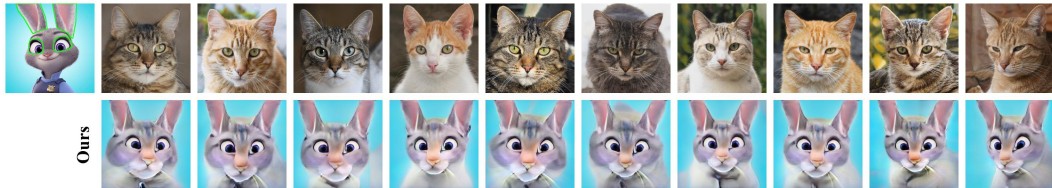

Figure 35: In the first row, the leftmost image is the reference, and the rest of the images are randomly generated by the source generator. The next row shows the adapted results of our method.

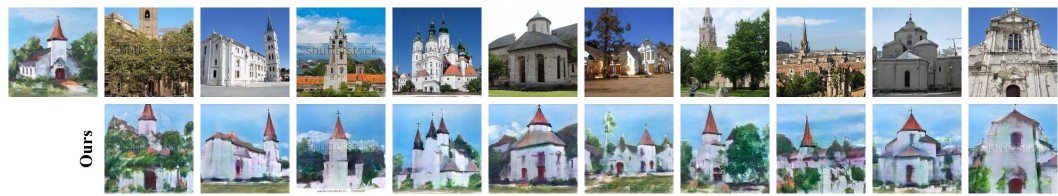

Figure 36: In the first row, the leftmost image is the reference, and the rest of the images are randomly generated by the source generator. The next row shows the adapted results of our method.

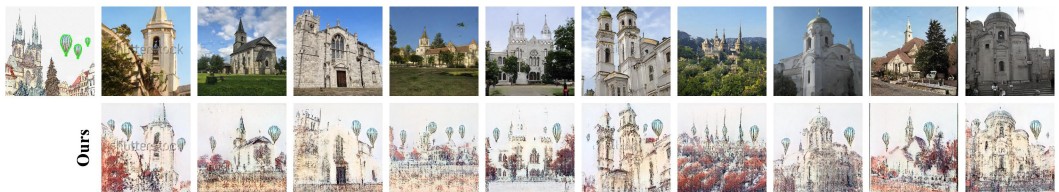

Figure 37: In the first row, the leftmost image is the reference, and the rest of the images are randomly generated by the source generator. The next row shows the adapted results of our method.

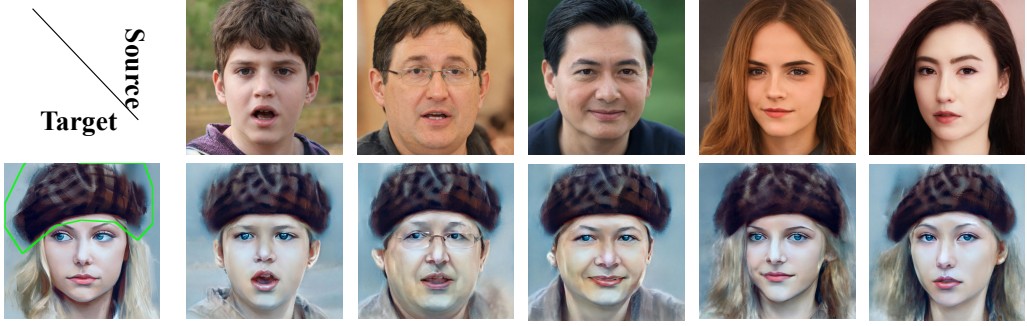

Figure 38: If the mask is not exact to cover the entity, our method can also generate reasonable results.