# OpenReview forum: "Generalized One-shot Domain Adaptation of Generative Adversarial Networks"
_NeurIPS.cc/2022/Conference — NeurIPS 2022 Accept_

### Official Review · Reviewer_Wx24 · 2022-07-09

**Rating:** 5
**Confidence:** 3
**Soundness:** 2 fair
**Presentation:** 2 fair
**Contribution:** 3 good

**Summary:**

This paper presents a method for one-shot domain adaptation, where a pre-trained GAN is leveraged to fine-tune the generator on one target image. First, given a source image, a latent vector is obtained by GAN inversion. Then, the generator is trained using a reconstruction loss, style and entity losses and a Laplacian regularizer, in order to transfer the style content of the target image into the source image while retaining the entity of the latter. Moreover, the proposed method admits a binary mask to select parts of the target image and to blend them into the source image, which is a novel contribution. Quantitative and qualitative results are presented when transferring from a few source images to a few target images from AAHQ dataset, as well as qualitative results for a few source and target images on AFHQ dog dataset and some source and target images on LSUN church dataset, showing a successful one-shot style transfer with optionally transferring objects from the target image.


**Questions:**

As it was written above, I would like to see averaged quantitative metrics over many source/target samples.


A sentence in l.176-177 claims that slice Wasserstein distance “leads to the same destination but with greater efficiency”, compared to the GAN loss. There is no reference cited nor I have seen an ablation experiment in the paper. Could the authors elaborate on this with empirical evidence or otherwise, remove the sentence altogether?

In l.35 it is mentioned that other methods produce artifacts when the entities are big. Could you point out where can this be observed in the figures?

How were the hyper parameters chosen? No discussion is provided with respect to that.



*Details on clarity*
* The task definition in l.106 is not clear and also fails to mention one of the most important details: the generator is fine-tuned on only one image from the target domain. As it stands, it appears as if a collection of target images were used instead.
* The methodology section focuses on one particular case, pretrained on FFHQ. Instead, the methodology should be general and only introducing the datasets and particular examples in the experimental section and Figure 2 as an illustration.
* Figure 2 is not clear, misses notation to quickly cross-reference it with the text. Moreover, the figure is not self-contained as there is some notation that is not explained in the caption. Additionally, this figure is referenced in the text before anything about the model is explained, which generates confusion.
* The explanation of the auxiliary net is lacking, specially to understand how the feature map $f_{ent}$ and mask $m$ are obtained. Is the mask predicted on the feature maps from the style-gan architecture? And if so, did you use a pre-trained U-Net or you trained one from scratch? How did you adapt a pre-trained U-Net in this case?
* Equation (1) misses $f_{ent}$ and $m$ as inputs to $aux$.
* l.187: “layers from pre-trained LPIPS” this is a experimental detail that does not belong in the methodology section.
* l.187: “upsampled by m” what does hat mean?
* Paragraph in lines l. 158-164 would be better suited in the experimental section.

* Where is the source of the images “Sketch, Disney and Arcane” that are mentioned in l.241? No reference is provided.



**Limitations:**

The limitations were discussed and the broader impact briefly touched upon.

**Strengths And Weaknesses:**

**Strengths**

The work is well motivated and the introduction of a binary mask to optionally import objects from the target image is a novel contribution. It targets a useful application and can be of use to the community and for content creators.

The usage of quantitative metrics to evaluate the results is a good addition to the user studies. Moreover, the two metrics used are reasonable in the context of face to face translation.

The methodology is sound and the claims are adjusted to the results presented in the experimental section.

**Weaknesses**

The paper lacks clarity and misses important information (more details in "Questions"). The related work is not well redacted and fails to situate the proposed method within other methods in the literature.

I would have expected the quantitative metrics to be computed and averaged across multiple samples for the target and source domains. Instead, those are only computed on hand-picked source-target images and risks providing only a biased positive view of the results.
The experimental setup is lacking  to properly quantitatively evaluate the method with respect to the baselines. Instead of computing them on a handful of samples, they should be computed over a dataset (or a randomly chosen subset of data) for the source and target domains in order to draw more robust conclusions.

---

> ### Author Response · Authors · 2022-08-02
> **Response to Reviewer Wx24 (Part 1/2)**
>
> Thanks for your careful review. We will explain your concerns in detail.
>
> __Q1. Averaged quantitative metrics over many source/target samples.__
>
> A1. Thanks for your advice. Since all the compared works (i.e., FSGA, MTG, JoJoGAN, and OSCLIP) only report the quantitative metrics on each source/target domain, we follow them to evaluate each source/target domain for a fair comparison. To get a more general conclusion, we conduct experiments on 50 target images including 25 images with entities and 25 images without entities. The average evaluation results are provided in the tables below.
>
> - The comparison of different methods. Here we do not evaluate the results of FSGA, since it suffers serious
>   over-fitting as depicted in Fig. 3 of the main paper. We can conclude our model performs better than previous works. For the
>   details and qualitative samples, please refer to Sec. 9 and Sec. 10 of the revised supplementary materials. The value
>   represents $mean_{std}^{ci}$, and $mean \pm ci$ denotes the confidence interval at 95\% confidence level.
>
>   | Metric | MTG  | OSCLIP | JoJoGAN | Ours |
>   | :------: | :-----: | :------: | :-------: | :----: |
>   | NME$\downarrow$    | $0.12^{0.01}\_{0.03}$ | $0.17^{0.04}\_{0.10}$   | $0.12^{0.03}\_{0.10}$    | $0.10^{0.04}\_{0.11}$ |
>   | ID$\uparrow$     | $0.16^{0.02}\_{0.06}$ | $0.19^{0.01}\_{0.03}$   | $0.17^{0.01}\_{0.04}$    | $0.27^{0.02}\_{0.05}$ |
>
> - We also provide the results of different weights of $\mathcal{L}\_{VLapR}$. The results prove that the regularization
>   is effective to preserve the source contents. For quantitative results please refer to Fig. 10 of the revised
>   supplementary materials.
>
>   | Metric | $\lambda\_{4}=0$ | $\lambda\_{4}=0.5$  | $\lambda\_{4}=2$ | $\lambda\_{4}=10$ |
>   | :------: | :----: | :------: | :-------: | :----: |
>   | NME$\downarrow$    | $0.11^{0.03}\_{0.10}$ | $0.10^{0.04}\_{0.11}$   | $0.10^{0.02}\_{0.06}$    | $0.09^{0.02}\_{0.06}$ |
>   | ID$\uparrow$     | $0.23^{0.02}\_{0.05}$ | $0.27^{0.02}_{0.05}$   | $0.30^{0.02}_{0.05}$    | $0.34^{0.02}_{0.04}$ |
>
> __Q2. The related work is not well-redacted.__
>
> A2: Thanks for your advice. We have reorganized the related work in the revised paper to make it better fit our article.
>
> __Q3. About slice Wasserstein distance in l.176-177.__
>
> A3:  In brief, SWD can completely capture the target distribution and computed fastly. The following explains why
> optimizing SWD is a more efficient and elegant way in our framework.
>
> - Firstly, we claim that both the style transfer and entity generation can be interpreted by learning the internal
>   distribution of example. As stated in l.171 to l.176, many works (_e.g._, SinGAN) adopt patch GAN loss to learn
>   internal distribution to generate new images. In theory, GAN losses essentially correspond to the divergence (vanilla
>   GAN loss versus JS-divergence) or distance (WGAN-gp loss versus Wasserstein distance) of the distributions. For style
>   transfer, the commonly used style losses, like gram loss and moment loss, are proved to align feature distributions of
>   stylized images and example [1]
>   .  [2] also shows that the more precise the alignment, the more faithful the stylization.
>
> - Then, although optimizing (patch) GAN loss is the most prevalent way to train the generative model, some drawbacks make it
>   inept for our framework. For example, since the discriminator of StyleGAN is very huge, FSGA (trained by GAN loss)
>   spent nearly 50 minutes per image and takes much more than 23Gib GPU memory. It also performs obvious mode collapse.
>   Besides, hence our core idea is to decouple the domain into style and entity, we can design specific losses for them.
>
> - Finally, SWD is very applicable to our framework in both theory and practice. As proved in [3,4], for distributions
>   $p$ and $q$, $p = q \Leftrightarrow SWD(p,q)=0$. The property makes it differ from and superior to the gram loss,
>   moment matching loss which cannot capture the complete distribution. In the experiment (See Sec. 2 in revised
>   supplementary materials), for style adaption, after careful adjustment the Gram and moment style loss have similar
>   performance to SWD in style adaption. However, for entity adaption they cannot work. Moreover, projecting the
>   high-dimensional data into the one-dimension space makes SWD superior in speed and memory cost. Our framework takes
>   about no more than 13GiB GPU memory and learns fast.
>
>   [1] Li, Yanghao et al. “Demystifying Neural Style Transfer.” IJCAI 2017.
>
>   [2] Kalischek, Nikolai et al. “In the light of feature distributions: moment matching for Neural Style Transfer.” CVPR
>     2021.
>
>   [3] Pitié, François et al. “N-dimensional probability density function transfer and its application to color
>   transfer.” ICCV 2005.
>
>   [4] Kolouri, Soheil et al. “The Radon Cumulative Distribution Transform and Its Application to Image Classification.”
>   IEEE Transactions on Image Processing 25 (2016): 920-934.

---

> > ### Author Response · Authors · 2022-08-02
> > **Response to Reviewer Wx24 (Part 2/2)**
> >
> > __Q4. Where can see other methods produce artifacts when the entities are big?__
> >
> > A4. Please see the last three cases in Fig.3 in the main paper. For other methods, the black hat pollutes their
> > synthesized hairs. The Zelda ornaments introduce obvious artifacts to the synthesized faces, and so does the mask. More
> > results can be seen in Fig. 22 to Fig. 31 of revised supplementary materials.
> >
> > __Q5. How were the hyperparameters chosen?__
> >
> > A5. We provide the details of parameter selection in the Sec.8 of the revised supplementary materials.
> >
> > __Q6. Details on clarity__
> >
> > A6. Thanks for your careful review. We will continue to revise and polish our paper according to your advice.
> >
> > 1. Task definition in l.106.
> >    A: We have modified the definition as "With the knowledge stored in a generative model $G_s$ pre-trained on source
> >    domain $\mathcal{S}$, a generative model $G_{t}$ is learned from $\boldsymbol{y}\_{ref}$ and $\boldsymbol{m}\_{ref}$
> >    to generate diverse images belonging to domain $\mathcal{T}$."
> >
> > 2. The methodology section focuses on one particular case. The methodology should be general.
> >    A: We heavily agree with your point that the methodology should be general. Actually, we follow previous works like
> >    MTG, Oneshot-CLIP, and JoJoGAN to take the face domain as the main example, which will be easy to understand for most
> > readers.
> >
> > 3. Figure 2 is not clear.
> >    A: We are sorry for this point, and working on ways to improve it. As the texts in the caption, we guide the readers
> >    to refer to Sec. 4.1. Although we do not describe the model too much in the caption, we think it has been clearly
> >    stated in Sec. 4.1 and self-consistent. We will continue to revise it.
> >
> > 4. About UNet and Eq. (1).
> >    A: Please note that texts from l.123 to l.35 are used to describe the networks. The aux adopts UNet architecture and
> >    will be trained from scratch. Eq. (1) is right since $m$ and $f_\
> >    {ent}$ are predicted by the UNet in aux. The architecture of UNet has been illustrated in Fig.4 of the revised
> >    supplementary materials.
> >
> > 5. The mean of “upsampled by m”.
> >    A: Since the size of $m$ is smaller than that of $y$, it should be upsampled to do the Hardmard product.
> >
> > 6. Images in l.241, no reference is provided.
> >    A: Thanks for your reminder. We have added the citations to them.

---

> > > ### Comment · Reviewer_Wx24 · 2022-08-08
> > > **Response to authors**
> > >
> > > Thank you for answers and extra experiments. Most of my concerns were addressed and I am raising the score accordingly.

---

### Official Review · Reviewer_5BYZ · 2022-07-10

**Rating:** 6
**Confidence:** 4
**Soundness:** 3 good
**Presentation:** 3 good
**Contribution:** 3 good

**Summary:**

This paper proposes a manifold regularized GAN adaption framework to deal with the generalized one-shot GAN adaption problem (i.e. the target domain to transfer contains both artistic styles and entities). To tackle this novel task, the paper modifies the architecture of the original generator by adding an additional auxiliary network to facilitate entity generation. To reduce the domain gap, they employ the sliced Wasserstein distance to minimize the divergence of the internal distributions between the exemplar and synthesis. Besides, they propose to use the variational Laplacian regularization $L_VlapR$ to mitigate the content distortion during training by preserving the geometric structure of the source manifold.


**Questions:**

See Weakness

**Limitations:**

Adequate

**Strengths And Weaknesses:**

The paper is well written and easy to follow. Both quantitative and qualitative experiments show that the proposed method has sufficient visual advantages over the competition. In addition, their image processing results look interesting, which I believe will be helpful for artistic creation.

My only concern is the definition of style. In my opinion, the geometric feature is also part of the style and not just the color. I feel the results for Disney and Zelda are not so successful. Domain adaptation should also include some geometric changes, but the Zelda results in the paper don't look like CG images, and the Disney results cannot keep the big eyes and exaggerated expressions.

---

> ### Author Response · Authors · 2022-08-02
> **Response to Reviewer 5BYZ**
>
> Thanks for your careful review and appreciation of our work. We will explain your concern in detail.
>
> __Q1. The definition of style. The geometric feature is also part of the style and not just the color.__
>
> A1: We agree with your point that the geometric feature is also part of the style. We will make a specific explanation
> about the concept of style in this paper, and show that by adjusting the weight of style, the geometric features can be
> changed in vision.
>
> - Since different people may have a different perceptions of style, it is really hard to put the exact concept of style
>   into words. Nonetheless, in the community of computer vision, the style of image is usually defined [1] to its __texture__, which can be represented by its internal statistics. By aligning the statistical features of the content image
>   with that of the style image, the content image will obtain the style. In our paper, we have also followed this rule
>   and used the internal distributions to describe the texture.
>
> - We consider that a successful and practical adapted model should preserve the user's identity. Thus our algorithm
>   not only transfers the style and entity knowledge, but also makes the contents of adapted syntheses faithful to that of the
>   source image, _i.e._, cross-domain correspondence. As shown in the figures of our paper and the supplementary materials, for most images the adapted models can generate joyful results and can be applied for editing faces. The user studies also prove the
>   point.
>
> - We provide an experiment in Fig. 9 of the revised supplementary materials, which shows increasing the weight of style loss can improve the performance of style effects with geometric change. Therefore, our method can control the degree of stylization by adjusting the style loss weight. And in our paper, we set it to a moderate value to preserve the content of the source image while adequately transferring the style.
>
> [1] Jing, Yongcheng et al. “Neural Style Transfer: A Review.” IEEE Transactions on Visualization and Computer Graphics
> 26 (2020): 3365-3385.

---

### Official Review · Reviewer_AEH2 · 2022-07-11

**Rating:** 5
**Confidence:** 4
**Soundness:** 3 good
**Presentation:** 2 fair
**Contribution:** 3 good

**Summary:**

This paper performs one-shot domain adaption of StyleGAN model and the model additionally supports adding new entities compared to previous methods. The proposed method is able to do so thanks to a handful of points. Among them, two techniques are interesting: 1) using sliced Wasserstein distance to compute the internal distribution distance between two images; 2) adding variant Laplacian regularization to alleviate cross-domain correspondence distortion.

**Questions:**

What is the reason to produce a mask using UNet since we can directly use the mask provided by users (if I understand correctly)? The underlying motivation is unclear. Plus, the authors should conduct an experiment to explain the benefit of doing it.

The paper missed comparison with other internal distribution distances like mse between gram matrices. To me, entity loss and style loss can be directly implemented with MSE between the gram matrices of deep features from two images. The authors are advised to try these loss functions to show the advantage of using sliced Wasserstein distance.

Why do you give the task of this paper a new name--generalized one-shot GAN adaption? From my understanding, the method additionally can add entities but that doesn't mean it is generalized.

**Limitations:**

The authors provide a section to talk about the limitations of this work. I agree with the authors about these limitations. I think one viable solution for the entity position problem is to inquire the users for an extra mask representing the desired location of the entity in the synthesized image.

**Strengths And Weaknesses:**

Strengths:

The experimental results are promising. Using sliced Wasserstein distance and variant Laplacian regularization make sense. The authors provide code and additional materials so the method seems reproducible. The learning speed of this approach is faster than previous methods.


Weaknesses:

The writing of this paper can be improved. From the abstract and introduction, it is hard to identify the contributions of the proposed approach. Fig. 2 is hard to follow. It seems weird to have Fig. 1 below the title.

---

> ### Author Response · Authors · 2022-08-02
> **Response to Reviewer AEH2**
>
> __Q1. The writing of this paper can be improved.__
>
> A1. Thanks for your careful review and advice on our work, we will polish our paper carefully. The last paragraph of the Introduction has been rewritten to highlight our contributions.
>
> __Q2, What is the reason to produce a mask using UNet since we can directly use the mask provided by users？__
>
> A2. Taking the StyleGAN pretrained on FFHQ as an example, we think about the "mask provided by users" could mean two
> things, the reference mask provided by users, or (most likely) the masks for the face images that the users want to
> stylize. To avoid misunderstandings, we will explain both of them.
>
> - Utilizing the reference mask provided by users as the masks for all syntheses. It is infeasible since the adapted
>   generator should synthesize the image with a reasonable entity that is in a proper location, of the proper shape, and
>   high-quality texture in vision. Fixing the mask will make most entities placed in the wrong location and of improper
>   shape.
> - Users provide the masks for their desired face images. It is a good idea for more accurate control and a viable
>   solution for our limitations. However, in this paper, we focus on adapting GAN. Since it is hard to manually get the
>   masks for all latent codes, we need to predict the masks by the unet.  As your say, we think that the masks provided by the user could be helpful when the results are not reasonable, or the user wants to specify the shape and position of the entity. In Sec.6 (Mask-guided transfer) of the revised supplementary materials, we provide the
>   experiment to show the feasibility of the idea.
>
> __Q3. Comparison with other internal distribution distances like mse between gram matrices.__
>
> A3. We will discuss it both theoretically and experimentally, and the conclusion is that Gram matrix loss
> can be used as Style loss, but cannot be used as entity loss.
>
> - In theory. As proved in [1], the Gram matrix loss is equivalent to the maximum mean discrepancy. For two distributions
>   $p$ and $q$, optimizing Gram matrix loss matches the mean statistics of $p$ and $q$, but $GramLoss(p,q)=0 \nRightarrow
>   p = q$. Some improved losses like BN statistics matching and moment matching [2] try to capture distribution more
>   exactly but still cannot capture the complete distribution. By contrast, [3,4] shows that $SWD(p,q)=0 \Leftrightarrow
>   p = q$. Hence, in theory, both Gram matrix loss and SWD can be applied to style transfer, but only SWD is suitable for
>   the generation task, where the internal distribution needs to be fully captured. For example, [5] proves that SWD can
>   be applied to simple texture synthesis by image-based optimization, while Gram loss cannot. Our works also prove that
>   combined with pretrained generators, SWD can be used to learn more complex images.
>
> - In the experiment. After careful adjustment of weights, we find the Gram loss can be used as style loss in our framework.
>   With the same vgg features as used in SWD, and weight 2e-6 for balancing the large Gram loss value, it gets similar
>   style adapted results like SWD in visual. We also try the moment matching [1], they do work very similarly. But for
>   entity adaption, taking all these style losses fail to generate the entities. Please refer to the Fig. 2 of the revised supplementary materials.
>
> [1] Li, Yanghao et al. “Demystifying Neural Style Transfer.” IJCAI 2017.
>
> [2] Kalischek, Nikolai et al. “In the light of feature distributions: moment matching for Neural Style Transfer.” CVPR 2021.
>
> [3] Pitié, François et al. “N-dimensional probability density function transfer and its application to color transfer.”
> ICCV 2005.
>
> [4] Kolouri, Soheil et al. “The Radon Cumulative Distribution Transform and Its Application to Image Classification.”
> IEEE Transactions on Image Processing 25 (2016): 920-934.
>
> [5] Heitz, Eric et al. “A Sliced Wasserstein Loss for Neural Texture Synthesis.” CVPR 2021.
>
> __Q4. Why do you give the task of this paper a new name--generalized one-shot GAN adaption?__
>
> A4. Firstly, in mathematics, if there exists a problem, the generalized problem means that it is more general, and the
> original problem is just a special case of the generalized problem. In our paper, the previous one-shot domain adaption
> is a special case when the mask is full-zero.
>
> Secondly, to our best knowledge, there has never been a study of entity transfer in either classic style transfer or
> the latest GAN adaption. Although only adding a mask of entity to the task settings, as shown in the paper, it will bring
> more interesting applications for artistic creations. Most importantly, it may bring more insightful eyes to consider
> how to better utilize the high-level knowledge restored in the pre-trained generators, rather than continuing to focus
> on the color or style transfer that is relatively mature nowadays. Hence, we gave the task a new name.

---

### Official Review · Reviewer_fun1 · 2022-07-14

**Rating:** 5
**Confidence:** 4
**Soundness:** 3 good
**Presentation:** 3 good
**Contribution:** 3 good

**Summary:**

This paper considers the one-shot domain adaption problem, it decoupled adaptation into two parts: style and entity transfer. Unlike most previous works that mainly focus on style transfer, the proposed method utilises binary entity mask with concise manifold regularized GAN design. The author modifies the architecture of the original generator to decouple the adaption of style and entity, and proposes the variant Laplacian regularization to smooth the network. Extensive experiments are conducted on various references with and without entities.


**Questions:**

How to get binary entity masks? The author just mentioned it in the abstract and fig.1, should add descriptions in the experiment part.

Why did this paper chose sliced Wasserstein distance?  Need more insight and theoretical depth.

In L139 'each w will be transformed into the style-fixed code...', how to do transform? Need more details.


**Limitations:**

The authors adequately addressed the limitations.

**Strengths And Weaknesses:**

This paper focuses on an interesting one-shot domain adaption problem, the idea of disentangling style and entity transfer is straightforward.

The motivation of this paper is to solve one-shot domain adaption problem. However, I have some concerns as follows:
1. About the problem setting: the paper only explores the domain adaption with a clear binary entity
mask on the target image, which does not always exist for the general one-shot domain adaption setting.

2. The disentangled GAN structure has been well studied by many existing works, it is hard for the reviewer to identify the real contributions of this paper.

3. The experiments are insufficient. The evaluation size is too small, the main result Tabel.1, only uses Fig. 3's images for quantitative evaluation.

4. Since the proposed method focuses on manifold regularization, some related Manifold GAN methods [1][2][3] are missing in comparison.
[1]MR-GAN: Manifold Regularized Generative Adversarial Networks
[2]MMGAN: Manifold-Matching Generative Adversarial Network
[3]Manifold-preserved GANs

---

> ### Author Response · Authors · 2022-08-02
> **Response to Reviewer fun1 (Part 1/3)**
>
> __Q1. About the problem setting: the clear binary entity mask does not always exist for the general one-shot domain
> adaption setting.__
>
> A1: Thanks for your careful reviews. Because there may exist different interpretations of the words _clear_ and _exist_. We
> will explain our generalized one-shot GAN task, and the entity mask in more detail.
>
> - About the task setting. Our task is generalized from the previous one-shot GAN adaption (OSGA) task. It not only focuses on (a) traditional style adaption like OSGA, but also (b) imports the entity from the target image to the syntheses.
> To achieve this goal, we introduce an extra mask to bring the two cases into a unified framework. When the users do not provide any mask, an all-zeros mask is computed automatically and the problem is solved as Case (a). Otherwise, both the style and the located entity will be transferred into the syntheses (Case (b)), which have never been studied and realized in previous works.
>
> - About the clear mask. We consider that the term “style” is a global concept to describe the distribution of color and texture in the target image, while the “entity” (like the hat) is a local concept about the specific object in the target image, which usually has explicit boundaries (e.g., the green lines in our paper). However, our method is robust to the mask and there is no need to provide a precise segmentation mask for the entity. A rough mask will still accomplish our purpose. For example, we locate the hat by a rough polygon, and the syntheses also look good (see Fig. 38 in Supplementary Materials).
>
> __Q2. How to get the mask?__
>
> A2: Since our target dataset contains only one image, obtaining its mask annotation is very efficient. In our
> paper, for the image containing the entity, we manually annotate the mask with the opensource tool LabelMe. We enclose the
> entity with lines that meet end to end, and the mask will be extracted automatically. The annotation usually takes no
> more than 1 minute. We think the increased labor cost is negligible compared with that of creating or looking for the desired
> target image. Moreover, the mask of the entity (e.g., a hat) can also be obtained by the pre-trained segmentation
> models.

---

> > ### Author Response · Authors · 2022-08-02
> > **Response to Reviewer fun1 (Part 2/3)**
> >
> > __Q3. Why use sliced Wasserstein distance?__
> >
> > A3: In brief, SWD can completely capture the target distribution and computed fastly. The following explains why
> > optimizing SWD is a more efficient and elegant way in our framework.
> >
> > - Firstly, we claim that both the style transfer and entity generation can be interpreted by learning the internal
> >   distribution of an example. As stated in l.171 to l.176, recent many works (e.g., SinGAN) adopt patch GAN loss to learn
> >   internal distribution to generate new images. In theory, GAN losses essentially correspond to the divergence (vanilla
> >   GAN loss versus JS-divergence) or distance (WGAN-gp loss versus Wasserstein distance) of the distributions. For style
> >   transfer, the commonly used style losses, like gram loss and moment loss, are proved to align feature distributions of
> >   stylized images and example [1]
> >   .  [2] also shows that the more precise the alignment, the more faithful the stylization.
> >
> > - Then, although optimizing (patch) GAN loss is the most prevalent way to train the generative model, some drawbacks make it
> >   inept for our framework. For example, since the discriminator of StyleGAN is very huge, FSGA (trained by GAN loss)
> >   spent nearly 50 minutes per image and takes much more than 23Gib GPU memory. It also performs obvious mode collapse.
> >   Besides, hence our core idea is to decouple the domain into style and entity, we can design specific losses for them.
> >
> > - Finally, SWD is very applicable to our framework in both theory and practice. As proved in [3,4], for distributions
> >   $p$ and $q$, $p = q \Leftrightarrow SWD(p,q)=0$. The property makes it differ from and superior to the gram loss,
> >   moment matching loss which cannot capture the complete distribution. In the experiment (See Sec. 2 in the revised supplementary
> >   materials), for style adaption, after careful adjustment the Gram and moment style loss have similar performance to
> >   SWD in style adaption. However, they cannot work for entity adaption. Moreover, projecting the high-dimensional data
> >   into the one-dimension space makes SWD superior in speed and memory cost. Our framework takes about no more than 13GiB
> >   GPU memory and learns fast.
> >
> >   [1] Li, Yanghao et al. “Demystifying Neural Style Transfer.” IJCAI 2017.
> >
> >   [2] Kalischek, Nikolai et al. “In the light of feature distributions: moment matching for Neural Style Transfer.” CVPR 2021.
> >
> >   [3] Pitié, François et al. “N-dimensional probability density function transfer and its application to color
> >   transfer.” ICCV 2005.
> >
> >   [4] Kolouri, Soheil et al. “The Radon Cumulative Distribution Transform and Its Application to Image Classification.”
> >   IEEE Transactions on Image Processing 25 (2016): 920-934.
> >
> > __Q4. How to get the style-fixed code?__
> >
> > A4: Please refer to Sec.4.2. We replace the style part (the latest $18-l$ vectors) of $\boldsymbol{w}$ by that
> > of $\boldsymbol{w}\_{ref}$ to get the style-fixed code $\boldsymbol{w}^{\sharp}$. The process is formulated by Eq. (3) in the main paper.
> >
> > $\boldsymbol{w}^{\sharp} = diag(\boldsymbol{\alpha}) \boldsymbol{w} + diag({1}-\boldsymbol{\alpha})
> > \boldsymbol{w}_{ref},\ \alpha\_{i}=\textbf{1}\_{i<=l}(i), \ i = 1,\dots,18. $
> >
> > $\textbf{1}$ is the indicative function, $diag$ is the diagonalization operator, and $l$ is a hyperparameter to control the
> > trade-off between content and style. Visualization results are provided in Fig.2 in the main paper, in which the yellow arrow brings the style part of
> > $\boldsymbol{w}_{ref}$ to $\boldsymbol{w}^{\sharp}$, and the syntheses will get the style of reference.

---

> > > ### Author Response · Authors · 2022-08-02
> > > **Response to Reviewer fun1 (Part 3/3)**
> > >
> > > __Q5. About The disentangled GAN structure and main contributions__
> > >
> > > A5. Although disentangled GAN structure has been well studied, we are the first to propose the meaningful
> > > disentanglement of style and entity for the GAN adaption. Our main contributions can be concluded into two aspects.
> > >
> > > 1) We generalize the one-shot GAN adaption (OSGA) task focusing on style transfer with the _entity adaption_. To our
> > >    best knowledge, there has never been a study of entity adaptation in either classic style transfer or the latest OSGA
> > >    field. This task is much more challenging since for each synthesis the entity should be in the right location, of
> > >    proper shape and have high-quality texture in visual. We believe this task could lead to more breakthroughs in the
> > >    field of the generative model, and bring more help to artistic creation.
> > >
> > > 2) In technique, we propose a novel and concise framework with the disentangled style and entity loss, and a manifold
> > >    regularization. The SWD for internal learning makes sense and speeds up the training tremendously. The regularization
> > >    is effective to avoid distortion by SWD. As mentioned by other reviewers, our method makes sense (Reviewer AEH2)
> > >    and sound (Reviewer Wx24).
> > >
> > > __Q6.The experiments are insufficient. The evaluation size is too small.__
> > >
> > > A6. Since the few-shot GAN adaption works like FSGA reports the metric results on each source/target domain, and
> > > one-shot adaption works MTG, JoJoGAN, OSCLIP only report user study results on each source/target domain, we follow them
> > > to report the results on each source/target domain for a fair comparison.
> > > To get a more general conclusion, we conduct experiments on 50 target images. The average evaluation results are
> > > shown in the tables below.
> > >
> > > - The comparison of different methods. Here we do not evaluate the results of FSGA, since it suffers serious
> > >   over-fitting as illustrated in our paper. We can conclude our model performs better than previous works. For the
> > >   details and qualitative samples, please refer to Sec. 9 and Sec. 10 of the revised supplementary materials. The value
> > >   represents $mean_{std}^{ci}$, and $mean \pm ci$ denotes the confidence interval at 95\% confidence level.
> > >
> > >   | Metric | MTG  | OSCLIP | JoJoGAN | Ours |
> > >   | :------: | :-----: | :------: | :-------: | :----: |
> > >   | NME$\downarrow$    | $0.12^{0.01}\_{0.03}$ | $0.17^{0.04}\_{0.10}$   | $0.12^{0.03}\_{0.10}$    | $0.10^{0.04}\_{0.11}$ |
> > >   | ID$\uparrow$     | $0.16^{0.02}\_{0.06}$ | $0.19^{0.01}\_{0.03}$   | $0.17^{0.01}\_{0.04}$    | $0.27^{0.02}\_{0.05}$ |
> > >
> > > - We also provide the results of different weights of $\mathcal{L}\_{VLapR}$. The results prove that the proposed regularization
> > >   is effective in preserving the source contents.  Please refer to Fig. 10 of the revised
> > >   supplementary materials for the qualitative results.
> > >
> > >   | Metric | $\lambda\_{4}=0$ | $\lambda\_{4}=0.5$  | $\lambda\_{4}=2$ | $\lambda\_{4}=10$ |
> > >   | :------: | :----: | :------: | :-------: | :----: |
> > >   | NME$\downarrow$    | $0.11^{0.03}\_{0.10}$ | $0.10^{0.04}\_{0.11}$   | $0.10^{0.02}\_{0.06}$    | $0.09^{0.02}\_{0.06}$ |
> > >   | ID$\uparrow$     | $0.23^{0.02}\_{0.05}$ | $0.27^{0.02}_{0.05}$   | $0.30^{0.02}_{0.05}$    | $0.34^{0.02}_{0.04}$ |
> > >
> > > __Q7. Comparison to other manifold GANs.__
> > >
> > > A7. We do not make the comparison, since Manifold GANs aim to fit the _real manifold_ of the large-scale dataset. In contrast,  for the one-shot task, the single image cannot formulate a manifold, which means that there does not exist a real manifold as the fitting target. Thus it cannot define the manifold concepts like radius or center in MMGAN. Our me method just utilizes the smoothness information of the source generator to preserve the relative relation invariant. As stated in line 207, we have discussed and compared our $\mathcal{L}\_{VLapR}$ with $\mathcal{L}\_{CDC}$ proposed in FSGA, which is a classic manifold regularization like Stochastic Neighborhood Embedding （SNE） in essence. The results show that our $\mathcal{L}\_{VLapR}$ has advantages in both theory and experiments for this task.

---

> > > > ### Comment · Reviewer_fun1 · 2022-08-09
> > > > **After Rebuttal**
> > > >
> > > > The author addressed most of my concerns. Thus, I tend to raise my score.

---

### Author Response · Authors · 2022-08-01
**Response to all reviewers**

We thank all the reviewers for their thoughtful reviews and constructive comments on our work! We are encouraged and
glad to hear the feedback from reviewers that:

1. Our work is well motivated (Wx24), well written and easy to follow (5BYZ).
2. Our proposed  __task__, _i.e._, generalized one-shot domain adaption, is a novel contribution (Wx24) and
   interesting (fun1). It targets a useful application to the community, and will be helpful to content creators (Wx24)
   and artistic creation (5BYZ).
3. Our proposed __idea__ that decouples the domain adaption into style and entity transfer is straightforward (fun1).
4. Our __method__ using sliced Wasserstein distance and variant Laplacian regularization makes sense (AEH2). It is also
   sound (Wx24) and faster (AEH2).
5. Our proposed __metrics__ are reasonable and a good addition to the existing user studies (Wx24).
6. The __results__ look promising (AEH2), interesting, and sufficient visual advantages over the competition (5BYZ).

As the advantages are comprehensive in various aspects, the reviewers have raised different questions, which
are quite helpful to improve the paper and dig into the task. We have addressed these questions with additional
experiments and clarifications, which have been added to the
updated paper and supplementary materials. In response to feedback, we provide individual responses below to address each reviewer’s concerns.

---

### Meta-Review · Area_Chair_abRZ · 2022-09-05

**Recommendation:** Accept
**Confidence:** Certain

**Metareview:**

This paper focuses on the one-shot domain adaption of GAN model. The idea of disentangling style and entity transfer is simple and effective. The meta-reviewer recommends acceptance of the paper, and the authors are encouraged to take the reviews into consideration when preparing a final version of the paper.

**Award:**

No

---

### Decision · Program_Chairs · 2022-09-14

Accept